



# Porewater $\delta^{13}C_{DOC}$ Indicates Variable Extent Of Degradation In Different Talik Layers Of Coastal Alaskan Thermokarst Lakes

Ove H. Meisel[1,2], Joshua F. Dean[1,2,3], Jorien E. Vonk[1,2], Lukas Wacker[4], Gert-Jan Reichart[2,5,6], A. Johannes Dolman[1,2]

[1]Department of Earth Sciences Vrije Universiteit Amsterdam, Amsterdam, 1081HV, The Netherlands
[2]Netherlands Earth System Science Center, Utrecht University, Utrecht, 3584CS, The Netherlands
[3]School of Environmental Sciences, University of Liverpool, Liverpool, L69 7ZT, UK
[4]Department of Physics, Ion Beam Laboratory, ETH Zürich, Zürich, 8093, Switzerland
[5]Department of Ocean Systems, NIOZ-Royal Netherlands Institute for Sea Research, Den Hoorn, 1797SZ, The Netherlands
[6]Department of Earth Sciences, Faculty of Geosciences, Utrecht University, Utrecht, 3508TA, The Netherlands

*Correspondence to*: Ove H. Meisel (o.h.meisel@vu.nl)

**Abstract.** Thermokarst lakes play an important role in permafrost environments by warming up and insulating the underlying permafrost. As a result, thaw bulbs of unfrozen ground (taliks) are formed. Since these taliks remain perennially thawed, they are zones of increased degradation where microbial activity and geochemical processes can lead to increased greenhouse gas emissions from thermokarst lakes. It is not well understood though to what extent the organic carbon (OC) in different talik layers below thermokarst lakes is affected by degradation. Here, we present two transects of short sediment cores from two thermokarst lakes on the Arctic Coastal Plain of Alaska. Based on their physiochemical properties two main talik layers were identified. A 'lake sediment' at the top with low density, sand and silicon content but high porosity. Underneath a 'deeper talik' (former permafrost soil) of high sediment density and rich in sand but lower porosity. Loss on ignition (LOI) measurements show that the organic matter (OM) content in the 'lake sediment' of $28 \pm 3$ wt% ($1\sigma$, $n = 23$) is considerably higher than in the underlying 'deeper talik' soil with $8 \pm 6$ wt% ($1\sigma$, $n = 35$), but dissolved organic carbon (DOC) leaches from both layers in high concentrations: $40 \pm 14$ mg/l ($1\sigma$, $n = 22$) and $60 \pm 14$ mg/l ($1\sigma$, $n = 20$), respectively. Stable carbon isotope analysis of the porewater DOC ($\delta^{13}C_{DOC}$) showed a relatively wide range of values from -30.74 ‰ to -27.11 ‰ with a mean of $-28.57 \pm 0.92$ ‰ ($1\sigma$, $n = 21$) in the 'lake sediment', compared to a relatively narrow range of -27.58 ‰ to -26.76 ‰ with a mean of $-27.59 \pm 0.83$ ‰ ($1\sigma$, $n = 21$) in the 'deeper talik' soil (one outlier at -30.74 ‰). The opposite was observed in the soil organic carbon (SOC), with a narrow $\delta^{13}C_{SOC}$ range from -29.15 ‰ to -27.72 ‰ in the 'lake sediment' ($-28.56 \pm 0.36$ ‰, $1\sigma$, $n = 23$) in comparison to a wider $\delta^{13}C_{SOC}$ range from -27.72 ‰ to -25.55 ‰ in the underlying 'deeper talik' soil ($-26.84 \pm 0.81$ ‰, $1\sigma$, $n = 21$). The wider range of porewater $\delta^{13}C_{DOC}$ values in the 'lake sediment' compared to the 'deeper talik' soil, but narrower range of comparative $\delta^{13}C_{SOC}$, along with the $\delta^{13}C$-shift from $\delta^{13}C_{SOC}$ to $\delta^{13}C_{DOC}$ together indicates increased stable carbon isotope fractionation due to ongoing processes in the 'lake sediment'. Increased degradation of the OC in the 'lake sediment' relative to the underlying 'deeper talik' are the most likely explanation for these differences in $\delta^{13}C_{DOC}$ values. As thermokarst lakes can be important greenhouse gas sources in the Arctic it is important to better understand the degree of degradation in the individual talik layers as an indicator for their potential in greenhouse gas release. Especially, as predicted warming of the Arctic in the coming decades will likely increase the number and extent (horizontal and vertical) of thermokarst lake taliks.



## 1 Introduction

Thermokarst lakes are common in permafrost landscapes of Siberia, Canada and Alaska where they regionally cover more than 40% of the land area (Lehner and Döll, 2004). Near Utqiaġvik, Alaska they cover 23% of the land surface (Engram et al., 2018). The name thermokarst lake refers to the process of lake formation through thaw-induced permafrost degradation resulting in the formation of topographic depressions in previously stable ground. This process is especially effective when the permafrost soils contain high proportions of ground ice (West and Plug,

2008). Along the Beaufort Sea coast of northern Alaska, permafrost soils locally contain up to 50% ice in soil volume (Kanevskiy et al., 2013). Upon thaw, these soils undergo substantial volume loss, causing soil collapse and large-scale subsidence. Thawing is further exacerbated through water bodies such as ponds which form in hydrologically closed depressions. The pond water raises the soil temperature of the underlying permafrost as dark water surfaces trap heat due to their low albedo. With incipient thawing, ponds can eventually grow to large

thermokarst lakes through horizontal and vertical permafrost degradation over time (West and Plug, 2008). Sudden processes like lake drainage, and flooding of drained lake basins, can also abruptly lead to the formation of new thermokarst lakes within days or weeks (van Huissteden et al., 2011), subjecting the permafrost landscape to constant change.

Thermokarst lake systems also play a vital role in the overall greenhouse gas budget and carbon balance of the

Arctic. Depending on their depth, location in the landscape, adjacent soil properties, levels of shoreline erosion, and local climate conditions, thermokarst lakes were net carbon sinks (Walter et al., 2006, 2007) or sources of atmospheric carbon (Anthony et al., 2014) during the Holocene. Favorable conditions for carbon sequestrations in these lakes are met when the input of organic material through shoreline erosion outpaces its degradation rate. Sediment accumulation rates can also increase when aquatic productivity is enhanced through supply from nutrient-

rich permafrost soil erosion (e.g. yedoma) or when the lake itself provides the physical conditions (e.g. anoxia) for carbon accumulation and preservation in deeper water (Anthony et al., 2014). Thermokarst lakes can act as net carbon sources when conditions for carbon sequestration are not met. Even conservative estimates expect an increase in future greenhouse gas emissions from thermokarst lakes through processes such as lake expansion due to warming air temperatures in the Arctic (van Huissteden et al., 2011).

Thermokarst lakes that freeze all the way to the lakebed in winter are referred to as bedfast-ice lakes. Once a thermokarst lake reaches a critical water depth which exceeds the maximum lake ice thickness in winter, the deepest part of the water column remains unfrozen year-round (Arp et al., 2012). These so-called floating-ice lakes then convey heat to the lakebed, as a result a thaw bulb or talik (a zone of permanently unfrozen ground) forms that first extends into the frozen 'lake sediment' and eventually propagates into the frozen soils below (Heslop et al., 2015).

The uppermost part of the talik consists of materials accumulated through lake sedimentation that steadily grow in thickness through ongoing deposition. The main sediment sources in thermokarst lakes are allochthonous material of terrestrial origin from erosion of permafrost shorelines (input of previously frozen SOC) and tundra vegetation as well as autochthonous material from aquatic productivity (Roiha et al., 2016). The deeper part of the talik contains the previously frozen ground of the permafrost underlying the lake.

Hugelius et al. (2014) estimated the total amount of SOC in the upper 3 m of the permafrost landscapes to 1307 Pg (± 170 Pg) of which 822 Pg were attributed to perennially frozen soils. Taliks can reach 95 m vertically into the permafrost under deep thermokarst lakes in Siberia (Schwamborn et al., 2000), but their growth is restricted and halts when a state of thermal equilibrium is reached (Boike et al., 2015). Lake taliks in the Arctic Coastal Plain of Alaska do not grow as deep and reach maximum depths of 53 m in ground ice-rich permafrost (Ling and Zhang, 2003). As

talik formation below thermokarst lakes progresses vertically, it introduces previously frozen SOC into the active carbon cycle by exposing it to year-round microbial degradation. In the same way these lakes function as natural incubators for permafrost OC that is eroded horizontally from its shorelines as talik expansion also occurs laterally (Walvoord et al., 2019). This shoreline OC becomes part of the lake sediment after deposition on the lakebed. The water-saturated, thawed sediments leach DOC, a readily bioavailable form of OC, into their porewater where it is





taken up and consumed by microorganisms (Nelson and Wear, 2014; Silveira, 2005; Spencer et al., 2015; Vonk et al., 2013). OC that is part of a talik as SOC or DOC is exposed to increased levels of degradation in comparison to the frozen grounds of the surrounding landscape (Heslop et al., 2015).

Despite extensive research on the importance of thermokarst lakes in the carbon cycle of permafrost regions, the role of taliks in thermokarst lake systems is not well understood. It is known that thermokarst lake taliks include thawed
permafrost soil in its deeper layers and a steadily growing layer of lake sedimentation at the top of the talik. These layers can differ significantly in OC content due to the different formation processes. In particular, it is unclear to what degree the uppermost talik layer of the 'lake sediment' and the 'deeper talik' layers are affected by degradation processes (Heslop et al., 2015). Here we use DOC concentrations, soil organic matter (SOM) content (based on LOI), as well as $\delta^{13}C_{SOC}$ and $\delta^{13}C_{DOC}$ ratios to evaluate the degree of degradation processes in different lake talik
layers. We analyzed these parameters in sediment cores from two thermokarst lakes near the city of Utqiaġvik, Alaska (Fig. 1). The different talik layers were characterized using physical and sedimentary techniques (X-ray fluorescence scanning, sediment density, water content, grain size, magnetic susceptibility, core imaging). The changes from $\delta^{13}C_{SOC}$ to $\delta^{13}C_{DOC}$ in the individual talik layers were of particular interest to determine whether OC degradation occurs at different rates in the individual carbon pools of the 'lake sediment' versus the 'deeper talik'.

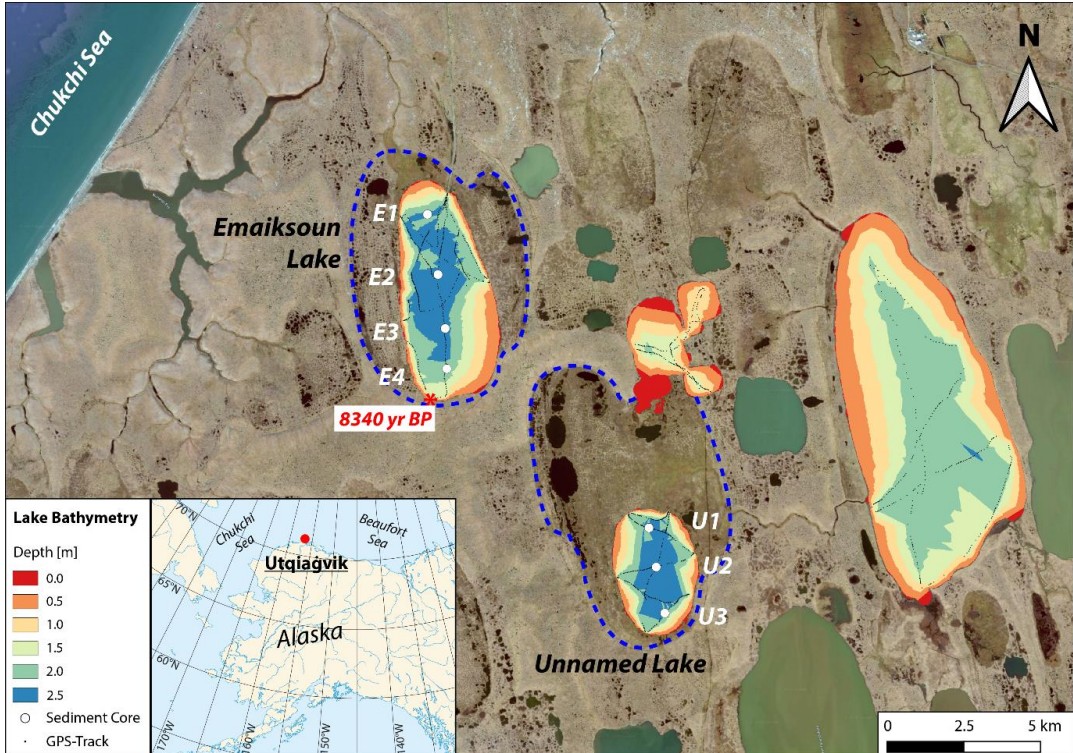


*Figure 1. Study area 5 km south of Utqiaġvik (Alaska) with lake bathymetry maps. Emaiksoun and Unnamed Lake were sampled for sediment core transects E1-E4 and U1-U3, respectively. Blue, dashed outlines indicate their remnant shorelines of previous lake extents (or old partially drained lake basins). The location and age of a radiocarbon shoreline sample is displayed at the southern shoreline of Emaiksoun Lake. (The overview map was created with the Generic Mapping Tool 6.1.1.*
*The main map was created in QGIS 3.16.0 with the OpenLayers Plugin through imported satellite images from © Google Maps [Map data: 2020 TerraMetrics]).*



## 2 Methodology

### 2.1 Field Site and Sampling

Sediment cores were collected along two transects at two thermokarst lakes on the Arctic Coastal Plain of Alaska, 4 and 8 km south of Utqiaġvik (formerly Barrow), the northernmost settlement in Alaska. The landscape of the Arctic Coastal Plain is characterized by a flat topography of 0 - 300 m above sea level with only minor elevation changes especially in the study area along the Chukchi Sea coastline (Sellmann et al., 1975). Utqiaġvik has a cold maritime climate with a mean annual temperature of -12 ˚C and an annual precipitation of 106 mm of which 63 % falls as rain during the summer months (Hinkel et al., 2003).

Emaiksoun and Unnamed Lake were selected for sediment coring based on their depths of more than 2 m (Fig. 1) in the bathymetry maps which classifies them as floating-ice lakes on the Alaskan Coastal Plane. Also Hinkel et al. (2012) and Engram et al. (2018) identified Emaiksoun Lake as floating-ice lake with comparable lake depth measurements of more than 2 m. Unnamed lake to the southeast of Emaiksoun Lake has a similar maximum depth of 2.5 m and can thus also be considered a talik-forming, floating-ice lake as the maximum winter ice thickness on the
Arctic Coastal Plain averages around 1.5 - 2 m (Hinkel et al., 2012).

The prevailing easterly and westerly winds in the region cause wave formation and shoreline erosion perpendicular to the wind direction that lead to North-South-oriented and elongated shapes of the lakes (Black and Barksdale, 1949; Livingstone, 1954) characteristic for this region (Fig. 1). These windy conditions also cause constant suspension and mixing of the uppermost sediment layers in the shallow lakes as most lakes do not exceed a water
depth of 2 m.

Geologically the Arctic Coastal Plain of Alaska is predominantly underlain by fluvial and marine deposits of Pliocene to late Pleistocene age. These deposits are attributed to the geological Gubik Formation which is divided into several subunits of different age and sedimentary characteristics (Black, 1964). The study area is underlain by one of these subgroups called Barrow Unit. It is described as a heterogeneous formation of fluvial, lagoon, beach,
lacustrine and shallow marine deposits that was deposited as recently as 120,000 years ago in the late Pleistocene and was frequently reworked by different agents of transport (marine, coastal, fluvial, lacustrine erosion and redeposition) especially throughout the Holocene (Black, 1964; Repenning, 1983).

A lake survey took place from 28 July - 1 August 2015 which included bathymetry measurements of four thermokarst lakes and shoreline sediment sampling. For the lake depth measurements, a Humminbird PiranhaMAX
160 fish finder was operated from a boat to collect depth data across the lakes. The spacing between GPS positions of the depth data is displayed in Fig. 1. Based on GPS data and depth measurements bathymetric lake maps were compiled in QGIS 3.16.0 by creating shoreline shapefiles and interpolated raster layers of the data points by applying the Triangulated Irregular Network (TIN) method from the build-in interpolation plug-in.

During a second field trip from 2 - 15 November 2015 a total of 14 short sediment cores (60 cm - 84 cm) were taken
from seven coring sites along north-south lines from two thermokarst lakes (Fig. 1; Table 1) with a manual, hand-held UWITEC gravity coring device. A transect of four coring sites from Emaiksoun Lake (E1 - E4) and another transect of three coring sites from Unnamed Lake (U1 – U3) were collected through holes drilled in the frozen lake surfaces. At each of the seven coring sites two sediment cores were collected approximately 1.5 m apart. One core from each site was used for sedimentary and geochemical analysis, the other for DOC porewater extraction. Layer
thicknesses between the two sediment cores of each pair deviated on average only 3 ± 2 cm. This suggests stable sediment deposition rates (locally), no major sediment deformation during coring and homogenous core properties at the same core depths for the paired cores across the short distances. In the field the sediment cores were stored horizontally in insulated boxes after sampling to keep them above freezing temperature (4 – 7 ˚C). Porewater samples for DOC analysis were extracted from one sediment core at each sampling location at a 10 cm resolution at
the end of the sampling day, maximally 6 hours after core collection. Sampling was carried out with 5 cm long



ceramic SMS rhizons (0.12 - 0.18 µm pore size; Rhizosphere Research Products B.V.) which were inserted into the sediment cores through small holes drilled after the coring. Pre-evacuated vials (9 ml) were attached to the rhizons to extract the desired porewater volume of at least 5 ml, at times that process took a few minutes but could last up to 12 hours depending on the soil properties and porewater content at the sampling depth. By running through the pores of

the rhizons the water samples were directly filtered of all particulates and microorganisms in the porewater, while still capturing virtually the entire DOC pool (Dean et al., 2018). The samples were subsequently also treated with 3 - 4 drops of potassium iodide (KI) to further prevent microbial activity. After extraction, the porewater samples and all sediment cores were stored at 4 ˚C in the insulated boxes used in the field, shipped in cooled containers (4 ˚C) and after 2 weeks of transit stored horizontally, dark and refrigerated at 4 ˚C at the laboratories of the Vrije Universiteit

Amsterdam.

| Lake | Core | Sampling Date | Lon [°] | Lat [°] | Core Length [cm] |
|---|---|---|---|---|---|
| Emaiksoun Lake | E1 | 9-nov-2015 | -156.77894 | 71.24838 | 76 |
| | E2 | 9-nov-2015 | -156.77586 | 71.24270 | 63.2 |
| | E3 | 9-nov-2015 | -156.77375 | 71.23761 | 85.5 |
| | E4 | 10-nov-2015 | -156.77326 | 71.23381 | 80 |
| Unnamed Lake | U1 | 12-nov-2015 | -156.71400 | 71.21877 | 68 |
| | U2 | 11-nov-2015 | -156.71189 | 71.21506 | 82 |
| | U3 | 11-nov-2015 | -156.70938 | 71.21071 | 86.5 |
| Shoreline sample | / | 27-jul-2015 | -156.77823 | 71.23098 | / |

*Table 1. Overview of sediment cores and shoreline sample details.*

### 2.2 Core Scanning Analysis

Core splitting, linescan core imaging and X-ray fluorescence (XRF) scanning at a 1 cm resolution with an XRF-Scanner (Avaatech Analytical X-Ray Technology) were carried out at the Royal Netherlands Institute for Sea Research (NIOZ). The XRF data were taken at X-ray intensities of 10 kV and 30 kV (Hennekam et al., 2019), measuring amongst other iron (Fe), calcium (Ca) and silicon (Si). At the Vrije Universiteit Amsterdam the cores were measured for magnetic susceptibility and gamma-ray bulk density with a GEOTEK Multi-Sensor Core Logger

(MSCL) at a 1 cm resolution (Witak et al., 2005). For both MSCL measurements the results were calibrated against standards of known density and magnetic susceptibility.

### 2.3 Sediment Analysis

Grain size analyses were carried out with a laser-diffraction particle sizer (Sympatec HELOS KR) at the sediment laboratory of the Vrije Universiteit Amsterdam. In total 58 samples were analyzed at a standard resolution of 10 cm,

core sections with particularly homogenous core properties (E4, 24 - 78 cm) were individually sampled at lower resolution (maximum distance of 23 cm between samples).

All cores were subsampled at a higher resolution (4 cm) across the boundary of 'lake sediment' and former permafrost soil of the 'deeper talik' to record the changes in grain size properties in higher resolution, an approach that was applied to all other core analysis. The grain size samples were brought to boiling point together with 5ml of

30% hydrogen peroxide ($H_2O_2$) to oxidize all organic components in the sediment. Secondly, the organic-free samples were treated with 5ml of 10% hydrochloric acid (HCl) to remove all carbonates. In a last step 300 mg of $Na_4O_2P_7$ were added to the samples and, subsequently heated to the boiling point to break up particle conglomerates



(van Buuren et al., 2020). Loss on ignition (LOI) data were collected with Thermo-Gravimetric Analysis (TGA) carried out on a LECO TGA 701 at the Vrije Universiteit Amsterdam. The samples were heated up incrementally
from room temperature to 550°C for several minutes while the weights before and after combustion were measured until they stabilized (van Buuren et al., 2020). The loss on ignition at 550°C is here used as equivalent to the amount of SOM in the sediment. The water content of the sediment is based on the difference between wet and dry weights of the samples by loss in moisture after drying at 60°C over a period of at least 48 hours until a stable sample weight was reached (Ackroyd, 1957).

**2.4 Carbon Isotopes and DOC**

The $\delta^{13}C_{SOC}$ analysis was carried out at the stable isotope laboratory of the Vrije Universiteit Amsterdam with a Flash NC 1112 series Elemental Analyzer coupled to a Thermo Finnigan DeltaPlus XP Isotope Ratio Mass Spectrometer (IRMS). Prior to analysis, the samples were weighed into silver capsules and pre-treated by fumigation in a desiccator for 12 hours with fuming HCl (37%). Samples were folded and double-wrapped in tin capsules prior
to analysis and calibrated against internationally distributed $\delta^{13}C$ reference materials USGS24 (-16.05 ± 0.07 ‰ VPDB), USGS40 (-26.24 ± 0.04 ‰ VPDB) and USGS41 (+37.76 ± 0.05 ‰ VPDB) (Coplen, 1996; Qi et al., 2003).

The filtered DOC porewater samples showed signs of flocculation and precipitation of solid materials a few days to weeks after sampling. Microbial activity can be ruled out as cause due to the sample treatment with KI and the rhizon filtering process. Possible explanations are the interaction of porewater and its dissolved compounds with air in the
head space of the sample vials or flocculation as a result of reduced micro turbulences in the sample vials. The precipitates were measured for their OC contents and stable carbon isotope values to account for any OC that was taken from the DOC pool through flocculation processes.

In order to record the original amount of carbon that was present as DOC the porewater samples were refiltered (pre-combusted, 0.7 µm GF/F filters) to measure both DOC concentrations in the filtrate as well as the precipitated OC
collected on the filters as newly formed particulate organic carbon (POC). The combined mass of detected DOC in the porewater and POC on the filters was summed up as a total carbon mass to recalculate the original DOC concentration prior to flocculation. The final $\delta^{13}C_{DOC}$ values are composed of the combined $\delta^{13}C$ values of the DOC and POC based on their OC mass-based weighted average.

The dried filters with collected POC were placed in silver capsules, 25 µl MilliQ water was added to wet the sample
upon which it was treated with 25 µl of 1M HCL for 30 minutes. Another 50 µl 1M HCl were added for an additional 30 minutes before drying at 60 ˚C for three hours. The prepared samples were combusted to $CO_2$ and quantified for their carbon mass with a Carlo Erba NC 2500 elemental analyzer at the Systems Ecology laboratory of the Vrije Universiteit Amsterdam. Subsequent $\delta^{13}C$ analysis of the POC was carried out on a coupled Thermo Finnigan Delta Plus IRMS. The $\delta^{13}C$ measurements were calibrated against internationally recognized standards
USGS24 (-16.05 ± 0.07 ‰ VPDB), USGS40 (-26.24 ± 0.04 ‰ VPDB), USGS41 (+37.76 ± 0.05 ‰ VPDB), IAEA-CH-7 (-32.15 ± 0.05 ‰ VPDB) (Coplen, 1996; Qi et al., 2003) and in-house standards of glutamic acid (-24.08 ‰) and birch leaf (-27.74 ‰).

The DOC and $\delta^{13}C_{DOC}$ measurements were carried out at the KU Leuven, Belgium with an Aurora 1030W TOC Carbon Analyzer (OI Analytical) coupled to a Thermo Delta V Advantage IRMS (Dean et al., 2020; Morana et al.,
2015). The porewater samples were pre-treated with 10% phosphoric acid ($H_3PO_4$) to remove dissolved inorganic carbon (DIC). Samples were then purged with 5% sodium persulfate ($Na_2S_2O_8$) at 97°C to oxidize all DOC to $CO_2$. The $CO_2$ was conveyed to the analyzing units of the carbon analyzer and the IRMS with nitrogen ($N_2$) as carrier gas. The calibration of the $\delta^{13}C_{DOC}$ measurements was based on an IAEA-CH-6 (-10.449 ‰ ± 0.033 VPDB) and a sucrose standard (-26.99 ‰). The DOC concentrations were calibrated against the same isotope standards at an array of
different concentrations.





### 2.5 Radiocarbon Dating

The radiocarbon ($^{14}$C) dating was carried out at ETH Zürich, Switzerland on 10 samples, of which seven were macrofossils (brown moss leaves and stems from E3 and U2, 16 cm resolution). Three samples were composed of
bulk sediment; from the top of cores E3, U2 and a tundra surface sample from the southern, eroding shoreline of Emaiksoun Lake. The brown mosses were picked using sterile tweezers under a microscope from sieved (63 μm) and sodium hydroxide-soaked (5% NaOH) bulk samples. Sample preparation for the $^{14}$C analysis was carried out at the Vrije Universiteit Amsterdam. The three bulk samples were fumigated in a desiccator for three days with 37% fuming HCl at 60 ˚C, the same procedure was repeated with 5% NaOH to neutralize the sample again, followed by
drying. The macrofossil brown moss samples were treated with an acid-base-acid (ABA) procedure. The samples were reacting with 0.5M HCl for 1 hour at 60 ˚C, then soaked in 0.1M NaOH for 45 minutes and again soaked in 0.5M HCl for 1 hour at 60 ˚C. In between each soaking step and after the ABA procedure each sample was rinsed three time with Milli-Q water. Afterwards the brown moss samples were dried at 60 ˚C for 24 h.

The fumigated bulk samples were directly analyzed after combustion with a gas-ion-source-equipped AMS
(MICADAS) system at the Laboratory for Ion Beam Physics at the ETH Zürich (Wacker et al., 2013). The macrofossil samples were graphitized and a high-precision measurement performed (Wacker et al., 2010). Calibration of the radiocarbon ages to ranges of 2σ calendar years (yr) before present (BP) was carried out using IntCal20 (Reimer et al., 2020).

### 3 Results

All seven cores contain a basal layer of 'deeper talik' (Units A and B – former permafrost soils that are now unfrozen) with a layer of 'lake sediment' on top (Unit C). Both layers are part of the talik that formed under these thermokarst lakes. However, their properties differ due to their different origins and processes of formation. In the following, we present the physical and geochemical results that identify and distinguish these different layers.

### 3.1 Physicochemical Properties

The physical core descriptions in Figs. 2 - 3 show that the bottom half to two-thirds of all sediment cores are made up by sediments that can be attributed to that Barrow Unit of the Gubik Formation ('deeper talik') – Units A/B (Black, 1964). The grain size data show that all cores (except E4 and U3) have a layer of silty clay with a low average sand content (8 ± 6 %, 1σ, $n = 10$) at their base (Unit A). The dominating color in Unit A was identified as a very dark gray (2.5Y 3/0) on the Munsell Soil Color Chart (1988). Small mussel shells or shell fragments were
frequently present as well as scattered clusters of unidentified plant remains and small, well-rounded pebbles. Overall, Unit A is here identified as a shallow marine deposit from the Barrow Unit.

Unit B is present in all seven sediment cores, in E4 and U3 it makes up the base layer due to the lack of a Unit A. The grain size distribution in this unit is heterogeneous and ranges from sand (U3) to sandy loam (E3) and silty loam (U1). This unit is characterized by a strong variation in sand content due to some degree of mixing with the adjacent
Units A and C. The sand content ranges from 5.8 % (U2, 41 cm) up to 87 % (U3, 70 cm and 84 cm). Overall, the common distinguishing parameter across Unit B in all cores is the high average sand content (41 ± 24 %, 1σ, $n = 25$) compared to Unit A and C. The XRF data for silicon (Si) also coincide with the elevated sand content (Figs. 2 - 3). The dominant color of Unit B was determined as a dark grayish (2.5Y 4/2) to grayish brown (2.5Y 5/2). The color change from Unit A to B is clearly distinguishable as a sharp boundary in all cores, except from E1, where it appears
as a more gradual transition. Furthermore, Unit B contains well-rounded pebbles, ranging in size from a few mm to more than 1 cm in diameter, which are homogenously spread out across the unit. Mussel shell fragments were rare in this unit. Unit B is here identified as a coastal beach deposit of the Barrow Unit.

**275** *Figure 2. Physical property plots of cores E1 – E4 from Emaiksoun Lake show the talik subdivided into three units (A, B, C). The 'deeper talik' soil (A/B) is distinguishable from the overlying 'lake sediment' (C) especially through sharp boundaries in color, porewater content, grainsize, sediment density and silicon content. A vertical macrofossil radiocarbon age profile (red numbers) and a bulk sediment age (blue number) at the sediment surface are displayed as rounded values in core E3.*



*Figure 3. Physical property plots of cores U1 – U3 from Unnamed Lake show the talik subdivided into three units (A, B, C). The 'deeper talik' soil (A/B) is distinguishable from the overlying 'lake sediment' (C) especially through sharp boundaries in color, porewater content, grainsize, sediment density and silicon content. A vertical radiocarbon macrofossil age profile (red numbers) and a bulk sediment age (blue number) at the sediment surface are displayed as rounded values in core U2.*






The deposition of Unit A (shallow marine) and B (coastal, beach) in that order can be interpreted as a succession of marine regression deposits capturing a seaward moving coastline during the Pleistocene (Brigham-Grette and Carter, 1992). The porewater content across both units is at a low average of $26 \pm 10$ wt% ($1\sigma$, $n = 35$) with four major outliers in E1 (51 wt%, 74 cm), E2 (54 wt%, 61 cm), U2 (44 wt%, 62 cm) and U3 (50 wt%, 58 cm). The sediment density across both units does not drop below 1.36 g/cm³ (U2, 65 cm) and stands at an overall high average $\pm 1\sigma$ of 290 $1.94 \pm 0.2$ g/cm³ ($n = 327$) with a maximum density of 2.3 g/cm³ (E4, 34 cm). The magnetic susceptibility data (Figs. 2 - 3) also indicate elevated levels of magnetic particles in Units A/B (apart from E1 and E2) reflecting the abundance of mineral soil present in contrast to the overlying Unit C. The classification of Units A/B as marine deposits of the Barrow Unit within the Gubik Formation suggest that these sediments were originally not deposited through the means of lake sedimentation but were an integrated part of the permafrost ground before thermokarst 295 lake and talik formation took place at the sampling location.

A third unit, Unit C, was identified at the top of all seven sediment cores which differs strongly from the underlying Units A/B. The boundary between Unit B and C is marked by an abrupt color change to a very dark brown (10YR 2/2). The unstratified sediment of Unit C is a homogenous mud, dominated by clay or silty clay with a low average sand content ($5 \pm 5$ %, $1\sigma$, $n = 23$), except for one outlier in E3 (25 %, 15 cm). The change in grain size is reflected 300 in a sharp boundary to Unit B especially in E1, E2, E4 and U3. The porewater content increases strongly from the bottom ($47 \pm 2.7$ wt%, $1\sigma$, $n = 7$) to the water-saturated top of Unit C ($72 \pm 2.4$ wt%, $1\sigma$, $n = 7$). The average water content of $70 \pm 11$ wt% ($1\sigma$, $n = 23$) is considerably higher than in the Units A/B ($26 \pm 10$ wt%, $1\sigma$, $n = 35$). The sediment of Unit C is for a large part made up by plant detritus of terrestrial origin, mainly in the form of unidentified plant remains but also of roots, small leaves and stems among which brown mosses, *Sphagnum* and 305 *Betula nana* were identified. Furthermore, freshwater ostracod shells and shell fragments were present in small numbers. The sediment density drops significantly at the Unit B/C boundary to a Unit C average of $1.24 \pm 0.1$ g/cm³ ($1\sigma$, $n = 212$) which stands in a strong contrast to the combined Units A/B density average of $1.94 \pm 0.2$ g/cm³ ($1\sigma$, $n = 327$). The low sand content across Unit C ($5 \pm 5$ %, $1\sigma$, $n = 23$) is also reflected in the XRF-Si count that is negligible in comparison to the high levels of Unit B (Figs. 2 – 3). Additionally, the Fe/Ca-ratios in Figs. 2 - 3 show 310 that marine carbonate (represented by Ca) is still present throughout Units A/B indicated by the very low values as opposed to elevated Fe/Ca values in Unit C where terrestrial sources (represented by Fe) appear more dominant and carbonates are hardly present. Based on these physical properties Unit C is classified as a homogenous 'lake sediment' that contains large amounts of terrestrial plant detritus that differs strongly from the underlying 'deeper talik' layers of Units A/B.

The Units A/B were here identified as Pleistocene, marine deposits of the Gubik Formation which were formerly part of the permafrost soil. From here onwards, they are collectively referred to as 'deeper talik', the overlying organic-rich mud of Unit C as 'lake sediment'.

### 3.2 Talik Carbon Stocks

The 'deeper talik' and 'lake sediment' differ particularly in their OC characteristics as displayed in Fig. 4. The most 320 prominent difference between the 'deeper talik' and the 'lake sediment' is the SOM content, shown as LOI [wt%] at 550 ˚C. The 'lake sediment' is characterized by a high LOI of $28 \pm 4$ wt% ($1\sigma$, $n = 23$) in contrast to the combined LOI of the 'deeper talik' with $8 \pm 6$ wt% ($1\sigma$, $n = 35$). The high LOI values in the 'lake sediment' support the observation from the core descriptions where abundant amounts of plant detritus were present. The change in LOI at the Unit B/C boundary is sharp in all cores with exception of U1 and U2 where a more gradual transition zone 325 between the units formed.



Figure 4. Carbon plots of Emaiksoun and Unnamed Lake displaying profiles for loss on ignition (LOI at 550°C), porewater dissolved organic carbon concentration (DOC) and stable carbon isotope ratios of the sediment ($\delta^{13}C_{SOC}$) and porewater DOC ($\delta^{13}C_{DOC}$). A prominent jump in LOI values marks the B/C boundary.



The SOM (represented by LOI) in all sediment layers is the source material for DOC leaching into the sediment porewater where it can be further degraded (Kindler et al., 2011; McCullough et al., 2018; Peter et al., 2016).

Nonetheless, the pronounced difference between LOI in the 'deeper talik' and the 'lake sediment' is not reflected in the DOC concentrations. On the contrary, the average DOC concentration in the SOM-poor soil of Units A/B ($60 \pm 14$ mg/l, $1\sigma$, $n = 20$) is higher than in the SOM-rich 'lake sediment' ($40 \pm 14$ mg/l, $1\sigma$, $n = 22$). While the DOC concentrations in the 'deeper talik' remain at stable levels across the profile, the DOC concentration in the 'lake sediment' decreases steadily from its base ($48 \pm 12$ mg/l, $1\sigma$, $n = 7$) towards the sediment surface at 1 cm core depth

($25 \pm 5$ mg/l, $1\sigma$, $n = 7$).

The DOC$_{norm}$ data displayed in Fig. 5a are normalized DOC concentrations converted to their total carbon mass in [mg] per 1 cm core section (63.6 cm$^3$) at their respective sample depth taking into account the known sediment volume, sediment density and porewater content. Normalized DOC$_{norm}$ values [in mg] better display the total amount of DOC present in each individual layer than relative DOC concentrations [mg/l]. The same approach of carbon

normalization per 1 cm sediment core slice (63.6 cm$^3$) at the sampling depth was applied to the SOM$_{norm}$ [g] data in Fig. 5a which is based on the LOI (550 °C) data. The normalized and quantified SOM$_{norm}$ expressed in mass units is also representative of the amount of SOC in the ground as OM generally contains about 58 wt% OC (Christensen and Malmros, 1982). For Figs. 5 and 6 linear interpolations of the data were made to create matching depth data pairs. When DOC concentrations and SOM content are both normalized for their sediment properties (density, water

content) at the sampling point, one obtains a more accurate representation of the total OC present at a given depth. This normalization shows that DOC$_{norm}$ (Fig. 5a) in the 'lake sediment' is directly dependent on the availability of SOM$_{norm}$ ($R^2 = 0.40$, P<0.001). The same correlation (Fig. 5a) is found in the 'deeper talik' soil ($R^2 = 0.59$, P<0.001).

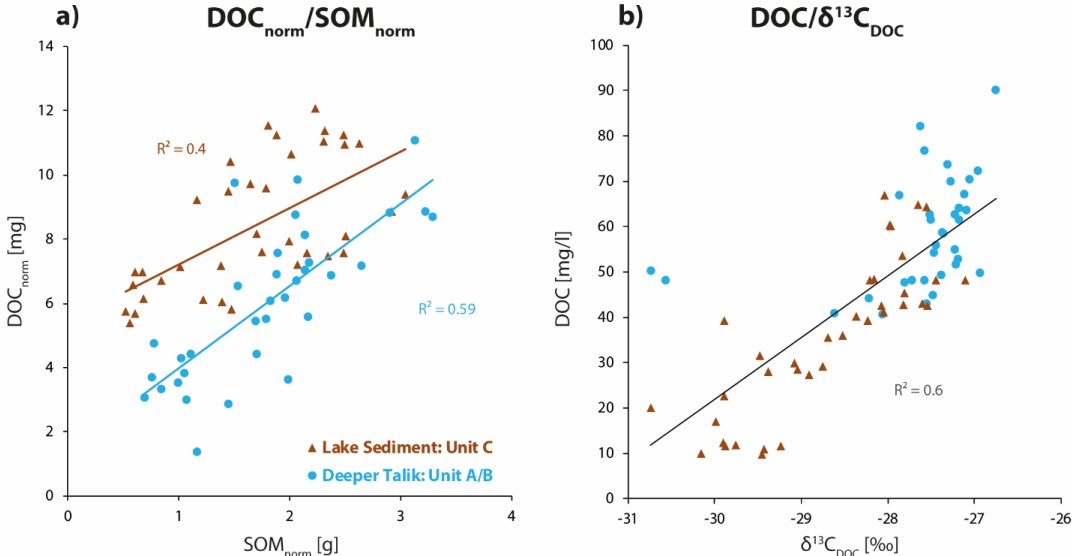

*Figure 5. a) DOC$_{norm}$ [mg] displays pore water DOC in mass units instead of concentrations. It shows that the total amount of DOC present in the 'lake sediment' is higher than in the 'deeper talik'. At the same time the total amount of SOM$_{norm}$ [g] ranges in the same order of magnitude in both layers. b) DOC concentrations are the highest in the 'deeper talik' while DOC concentrations in the 'lake sediment' have a broader concentration range. At the same time there is a trend across the complete talik profile of higher δ$^{13}$C$_{DOC}$ values with higher DOC concentrations.*




The higher the SOM content (and therefore SOC content) the more OC is leached into the sediment porewater as DOC. Furthermore, the normalized carbon plots (Fig. 5a) show that the total amount of OC available as DOC is in fact higher in the 'lake sediment' than in the 'deeper talik' soil. At the same time the total amount of OC available as $SOM_{norm}$ [g] is almost identical in both, the 'lake sediment' and the 'deeper talik'.

### 3.3 Stable Carbon Isotopes

The linear regression line in Fig. 5b shows a significant trend of higher $\delta^{13}C_{DOC}$ values with higher DOC concentrations across the complete talik profile ($R^2 = 0.60$, P<0.001). This finding confirms the observations from Fig. 4 where DOC concentrations were increasing and $\delta^{13}C_{DOC}$ values were higher with progressing talik depth. The trend of higher $\delta^{13}C_{DOC}$ values with depth is particularly pronounced within the 'lake sediment' from the top at 5 cm depth (-29.59 ± 0.68 ‰, 1σ, $n = 7$) to the bottom of the cores (-27.95 ± 0.59 ‰, 1σ, $n = 7$). Average $\delta^{13}C_{DOC}$ values of the 'deeper talik' (-27.59 ± 0.83 ‰, 1σ, $n = 21$) are significantly different from the 'lake sediment' (-28.57 ± 0.92 ‰, 1σ, $n = 21$), as confirmed in a one-way ANOVA test (P<0.001, F = 12, $F_{crit}$ = 4) which was performed as the values from the 'deeper talik' are not normally distributed. The average $\delta^{13}C_{SOC}$ values of the 'deeper talik' and 'lake sediment' are -26.84 ± 0.81 ‰ (1σ, $n = 21$) and -28.56 ± 0.36 ‰ (1σ, $n = 23$), respectively. Generally, $\delta^{13}C_{SOC}$ values across the 'lake sediment' slightly increase with depth from the top at 1 cm core depth (-28.91 ± 0.14 ‰, 1σ, $n = 7$) to the bottom of the unit (-28.25 ± 0.39 ‰, 1σ, $n = 7$). When plotting $\delta^{13}C_{DOC}$ and $\delta^{13}C_{SOC}$ against each other (Fig. 6a), two opposing trends stand out. Firstly, the $\delta^{13}C_{DOC}$ values in the 'deeper talik' soil ($R^2 = 0.24$, P<0.04) stay within a narrow range of 0.82 ‰ (min. -27.58 ‰ to max. -26.76 ‰, excluding one outlier at -30.74 ‰) while the $\delta^{13}C_{SOC}$ has a broad range of 2.17 ‰ (min. -27.72 ‰ to max. -25.55 ‰). The opposite relation ($R^2 = 0.44$, P<0.001) is observed for the 'lake sediment' where the $\delta^{13}C_{DOC}$ has a wide range of 3.63 ‰ (min. -30.74 ‰ to max. -27.11 ‰) while the $\delta^{13}C_{SOC}$ is limited to a narrow range of 1.43 ‰ (min. -29.15 ‰ to max. -27.72 ‰).

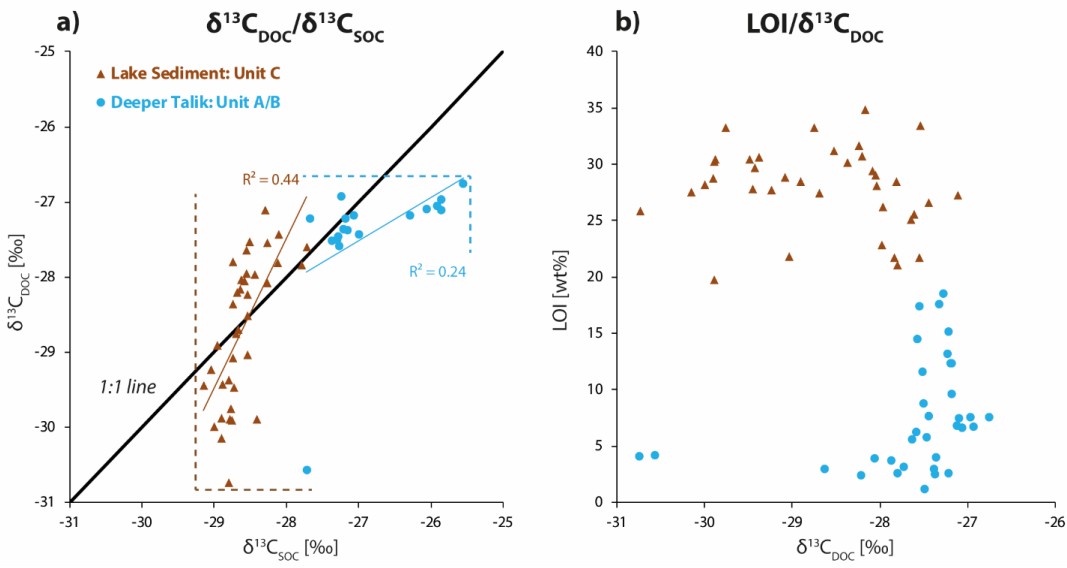

*Figure 6. a) The 'lake sediment' $\delta^{13}C_{SOC}$ value range is narrow while its $\delta^{13}C_{DOC}$ value range is wide, the opposite trend is the case for the 'deeper talik. The dashed lines highlight the respective $\delta^{13}C_{DOC}$ and $\delta^{13}C_{SOC}$ value ranges. The 1:1 line visualizes the changes in values from $\delta^{13}C_{SOC}$ to $\delta^{13}C_{DOC}$ in both layers. b) The loss on ignition (LOI) data of the 'lake sediment' are stable at a high level across both lakes and all cores while coinciding with a high variability in $\delta^{13}C_{DOC}$ values. Conversely, the $\delta^{13}C_{DOC}$ values of the 'deeper talik' stay in a narrower range independent of the variability in LOI values. LOI is here used as an equivalent for soil organic matter (SOM).*



The LOI data (equivalent for SOM content) are plotted against $\delta^{13}C_{DOC}$ in Fig. 6b, further supporting the observation
of different value ranges in $\delta^{13}C_{DOC}$ between the 'lake sediment' and the 'deeper talik' from Fig. 6a. It stands out that
the 'lake sediment' is more variable in its $\delta^{13}C_{DOC}$ range with 3.63 ‰ (min. -30.74 ‰ to max. -27.11 ‰) with a
range in LOI values of 15 wt% (min. 19.8 wt% to max. 34.8 wt%) in contrast to the underlying 'deeper talik' where
the range in LOI values is somewhat wider with 17.3 wt% (min. 1.2 wt% to max. 18.5 wt%) but the $\delta^{13}C_{DOC}$ values
are closer to each other with a range of only 1.87 ‰ (min. -28.63 ‰ to max. -26.75 ‰). The two $\delta^{13}C_{DOC}$ outliers in
the 'deeper talik' are both based on a single sample that resulted in a considerably lower $\delta^{13}C_{DOC}$ value than all other
samples from that unit (-30.74 ‰), the second outlier at -30.57 ‰ is the result of an interpolation with that data
point.

### 3.4 Radiocarbon Dating

All radiocarbon ages are listed in Table 2 and are displayed with rounded mean values in Figs. 1 - 4. One bulk
sediment $^{14}$C sample stems from the base of an eroding shoreline (Fig. 1, ~ 0.5 m above lake water level) and mainly
contains plant detritus and peaty soil. The uncalibrated $^{14}$C age at this site is 8340 ± 80 yr BP, demonstrating that
Early Holocene aged permafrost OC is readily available at the land surface and is actively eroded along the
shorelines of coastal Alaskan thermokarst lakes. Two more bulk sediment $^{14}$C samples were analyzed from the top
surfaces of core E3 with 3400 ± 70 yr BP (Fig. 2) and core U2 with 2785 ± 70 yr BP (Fig. 3). These bulk $^{14}$C ages
differ from another by 615 years but are in the same order of magnitude. The age difference between lake surface
sediment and eroding shoreline shows that despite the supply of older permafrost OC by erosion, large amounts of
contemporary carbon turnover must still take place in these lakes and contribute substantially to the overall lake
sedimentation.

| Core/ Location | Lab# ETH- | Depth [cm] | Material | $^{14}$C Age [yr BP] | Age Uncertainty [yr] | Calibrated $^{14}$C Age 2-σ range [cal. yr BP] |
|---|---|---|---|---|---|---|
| E3 | 97287 | 1 | Bulk sediment | 3,400 | 70 | / |
| | 97308 | 1 | Brown moss leaves and stems | 1,595 | 20 | 1,539 - 1,414 |
| Emaiksoun Lake | 97299 | 20 | | 1,285 | 20 | 1,281 - 1,181 |
| | 97307 | 33 | | 2,355 | 20 | 2,433 - 2,336 |
| U2 | 97288 | 1 | Bulk sediment | 2,785 | 70 | / |
| | 97306 | 1 | Brown moss leaves and stems | 775 | 20 | 729 - 675 |
| Unnamed Lake | 97304 | 20 | | 810 | 20 | 759 - 684 |
| | 97309 | 34 | | 1,230 | 20 | 1,258 - 1,073 |
| | 97297 | 50 | | 4,075 | 25 | 4,799 - 4,444 |
| Shoreline sample | 97296 | / | Bulk peaty soil sample | 8,340 | 80 | / |

*Table 2. Details of radiocarbon data from sediment cores E3, U2 and one shoreline sample. The calibrated age ranges were
calculated with IntCal20 (Reimer et al., 2020).*

In order to create age profiles of the lake sedimentation and determine the lake ages, macrofossil samples were
collected at the deepest and potentially oldest sections of the two lakes from the cores E3 and U2. The radiocarbon
samples of E3 from Emaiksoun Lake were exclusively collected in the 'lake sediment' and resulted in a mixed age
profile with the succession of 1539 - 1414 cal. yr BP (1 cm), 1281 - 1181 cal. yr BP (20 cm) and 2433 - 2336 cal. yr
BP (33 cm) (Table 2).

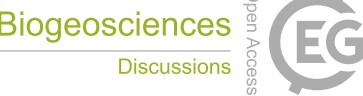

The average age difference of 1923 yr between the much older bulk sample (3400 ± 70 yr BP) and the macrofossil sample (1539 - 1414 cal. yr BP) at the top is noteworthy although both were taken at the same core depth of 1 cm (Fig. 2).

In contrast to E3 the [14]C age profile of U2 is chronological, the first three samples were also taken from the 'lake sediment', but here a fourth sample was analyzed from the 'deeper talik'. The succession of the calibrated age ranges in core U2 is 729 - 675 cal. yr BP (1 cm), 759 - 684 cal. yr BP (20 cm), 1258 – 1073 cal. yr BP (34 cm) and 4799 - 4444 cal. yr BP (50 cm) (Figs. 3 - 4). The [14]C ages from the 'lake sediment' in core U2 are considerably younger than in nearby Emaiksoun Lake. The bulk radiocarbon age and the macrofossil sample from the top of core U2 differ by 2083 yr resulting in a similar age difference as the equivalent sample pair in core E3.

## 4 Discussion

The 'deeper talik' layers were identified as marine deposits allocated to the Barrow Unit of the Gubik Formation in both lakes. They can therefore also be identified as part of the former permafrost ground which did not form through the means of lake sedimentation. With that distinction as premise, we can address our main question whether the relative degree of carbon degradation differs between the 'deeper talik' and the 'lake sediment' by comparison.

### 4.1 Lake Formation and Sedimentation

Radiocarbon dating of thermokarst 'lake sediment' is challenging for a number of reasons. Firstly, the lakes are shallow and often do not exceed a water depth of 2 m allowing for wind-induced wave action to affect the water column until the lakebed and thereby repeatedly suspend and redeposit the sediment which results in mixing of the uppermost sediment layer. This wave-induced mixing particularly applies to lakes in the flat, wind-exposed Arctic Coastal Plain of Alaska (Lenz et al., 2016). Secondly, the variable input of OC from shoreline permafrost erosion (old) and tundra vegetation (young) creates an inconsistent and mixed [14]C signal in the lake deposits. As a result of this continuous reworking, the 'lake sediment' is not reliably layered or varved which is a prerequisite for chronological radiocarbon profiles. The drastic age differences of 1923 yr (E3) and 2083 yr (U2) between the bulk sediment and macrofossil sample pairs from the tops of both cores (1 cm) are a good display of the sediment mixing and redeposition effects in thermokarst lakes as they show the dependency of [14]C ages on the origin of their source material (bulk sediment vs. macrofossils).

Although the exact chronology of the [14]C age profiles remain somewhat unclear due to mixing and reworking of the sediment (including macro fossils), it is clear that the lakes are several hundred to a few thousand years old. The calendar ages from the bases of the 'lake sediment' further suggest that Emaiksoun Lake (2433 - 2336 cal. yr BP, 33cm) is older than Unnamed Lake (1258 - 1073 cal. yr BP, 34 cm) as these basal ages indicate the onset of lake formation. Due to the mixing of old and recent OC in thermokarst lakes the true beginning of lake formation cannot be derived from these [14]C ages though. Potentially the lakes could be appreciably younger or older than indicated depending on the lateral inputs involved and changes therein through mixing processes. Even lake ages of only a few hundred years are not unusual in Arctic permafrost regions as these landscapes can undergo abrupt changes, for example, through sudden lake drainage and subsequent reflooding of nearby drained lake basins (Jones and Arp, 2015). The outlines of remnant shorelines of both lakes in Fig. 1 also show that they possibly had larger total lake surface areas in the past or refilled pre-existing drained lake basins.

In core U2 of Unnamed Lake, an additional sample was [14]C-dated in the 'deeper talik' of Unit B (50 cm depth). The measured age range of 4799 - 4444 cal. yr BP (Figs. 3 – 4, Table 2) is considerably older than the age at the base of the 'lake sediment' at 34 cm with 1258 - 1073 cal. yr BP. Nevertheless, the 'deeper talik' age range at 50 cm (Unit B) of 4799 - 4444 cal. yr BP is much younger though than the Pleistocene-aged Barrow Unit of the Gubik Formation (Repenning, 1983). This can be seen as an indication for partial reworking of the Pleistocene deposits at least





towards its top during the past 4799 - 4444 cal. yr BP and mixing with material from aquatic production, eroded shoreline permafrost or tundra vegetation.

465    **4.2 Carbon degradation**

The measured DOC concentrations of the 'deeper talik' are higher than in the 'lake sediment' (Fig. 4). The circumstance that the 'deeper talik' is of marine origin and contains Ca (Figs. 2 - 3), hence also carbonates, likely leads to a higher pH in the 'deeper talik' compared to the 'lake sediment'. It is known that SOM extractability is highly dependent on pH and that a higher pH enhances the solution of dissolved organic matter (Curtin et al., 1998, 470    2016). Consequently, the high DOC concentrations in the 'deeper talik' might be largely pH-driven. However, once DOC concentrations are normalized and expressed as $DOC_{norm}$ in mass units [mg], it becomes apparent that the total amount of DOC in the 'lake sediment' is actually higher than in the underlying layers of the 'deeper talik' (Fig. 5a). Interestingly, when normalizing SOM, both layers appear to contain equal amounts of SOM (Fig. 5a). In other words, both the 'lake sediment' and the 'deeper talik' layers contain approximately the same amount of SOM at a 475    given depth. However, in the 'lake sediment' the DOC leaches into the sediment porewater more effectively than in the 'deeper talik' (visible as higher DOC yields, Fig 5a), this might be due to the presence of younger more reactive SOC in the 'lake sediment' in comparison to the older 'deeper talik'.

In Fig. 5a we also show that the formation of porewater DOC is directly linked to the availability of SOM/SOC in the surrounding sediment, as yields of DOC (as $DOC_{norm}$) increase with increases in $SOM_{norm}$ content. This is the 480    case for the 'deeper talik' and the 'lake sediment' alike, showing that the process of DOC formation in the talik itself appears comparable throughout its individual layers (Fig. 5a). From these correlations we can also deduce that DOC formation in these thermokarst lake systems appears to mostly take place in situ in the talik through direct contact of porewater with adjacent SOC in the sediment and is less likely to be affected by vertical or lateral hydrological transport across or within the talik layers. Another feature common to both units is the trend of higher $\delta^{13}C_{DOC}$ values 485    with higher DOC concentrations (Fig. 5b). This is expressed in a notable correlation across all talik layers ($R^2 = 0.60$, P<0.001). The 'deeper talik' generally has considerably higher DOC concentrations and higher $\delta^{13}C_{DOC}$ values compared to the 'lake sediment' where DOC concentrations are lower and $\delta^{13}C_{DOC}$ values more negative (Fig. 5b).

The $\delta^{13}C_{SOC}$ values of the 'deeper talik' layers and the 'lake sediment' (Fig. 6a) differ considerably from one another, which also likely impacts the $\delta^{13}C_{DOC}$ signatures since SOC is the source material for in-situ DOC formation. The 490    differing $\delta^{13}C_{SOC}$ values in the two main talik layers could be an expression of the difference in the sediment source of the two layers. The 'deeper talik' layers are composed of reworked Pleistocene deposits which contain both terrestrial and marine components. The 'lake sediment' on the other hand consists mainly of eroded, organic-rich, terrestrial material from the surrounding tundra landscape. As Arctic plants are solely made up by $^{12}C$-enriched C3 plants the terrestrial $\delta^{13}C$ values in the 'lake sediment' are more negative than the marine-influenced $\delta^{13}C$ values in 495    the 'deeper talik' (Tieszen, 1973). In addition to a different source, the older sediment of the 'deeper talik' might have undergone degradation over a longer period of time than the younger 'lake sediment' further altering its $\delta^{13}C_{SOC}$ signature towards higher values (Blair et al., 1985). In sum, the difference in $\delta^{13}C_{SOC}$ values between 'lake sediment' and 'deeper talik' alone cannot be used to assess the level of degradation in the individual layers as their stable carbon isotope signatures might be largely source-driven. The changes from $\delta^{13}C_{SOC}$ to $\delta^{13}C_{DOC}$, however, are more 500    appropriate to assess degradation states as this transition involves microbial and biogeochemical processes. This assumption is also based on the factor that DOC is highly bioavailable, as it is quickly consumed by microorganisms and transformed by biochemical degradation (Battin et al., 2008) and thereby making it a more sensitive indicator for the degree of OC degradation.

Ågren et al. (1996) described how DOC degradation can lead to both, an increase and decrease in $\delta^{13}C_{DOC}$ values in 505    the remaining porewater DOC after degradation. Microbial degradation of OC usually coincides with an increase in $\delta^{13}C_{DOC}$ values as $^{12}C$ is preferentially used during microbial respiration and thereby leaves the remaining OC enriched in the heavier $^{13}C$ (Ågren et al., 1996; Blair et al., 1985). At the same time, biochemical degradation can



lead to more negative δ13C values in the remaining carbon pool. Preferential removal of the most labile carbon compounds such as simple polysaccharides which are naturally enriched in $^{13}C$ lead to an accumulation in less

degradable compounds like lignin which are naturally more depleted in $^{13}C$ (Ågren et al., 1996; Benner et al., 1987). Another process to potentially have an impact on $\delta^{13}C_{DOC}$ signatures is the selective sorption of specific DOC compounds to mineral surfaces in the soil. Certain DOC compounds can be naturally enriched (carboxyl groups) or depleted (lignin, lipids) in $^{13}C$ and their preferential removal through sorption can change the $\delta^{13}C_{DOC}$ value of the remaining DOC pool depending on the compositional change (Kaiser et al., 2001). Based on the assumption that the

sum of all these degradation processes leads to an overall more variable range in $\delta^{13}C_{DOC}$ values the values of the 'lake sediment' may be interpreted as an indicator for ongoing degradation. The DOC samples from the 'deeper talik' appear to be less affected by (recent) degradation as their $\delta^{13}C_{DOC}$ values stay within a considerably narrower range as visualized in Fig. 6a. This could also be related to the older age of this layer, likely leading to a relative larger proportion of recalcitrant compounds remaining upon long-term in-situ degradation of OC. In contrast, the

'lake sediment' is replenished with $^{13}C$-depleted fresh terrestrial OM as illustrated by an average core top $\delta^{13}C_{SOC}$ value of -28.91 ± 0.14 ‰ (1σ, $n = 7$).

Which mechanisms cause the limited variability in $\delta^{13}C_{DOC}$ values in the 'deeper talik' remains uncertain. Likely, sorption processes would be expected to have the strongest impact in the 'deeper talik' due to its mineral-rich soil and relative abundance in clay particles which provide high mineral surface areas for DOC sorption. An increase in

$\delta^{13}C_{DOC}$ values relative to $\delta^{13}C_{SOC}$ can be caused by the preferential sorption of isotopically lighter and hydrophobic compounds to mineral surfaces (Kaiser et al., 2001). But based on our data, the process of sorption does not appear to outweigh changes in $\delta^{13}C_{DOC}$ through microbial degradation as most of the points in Fig. 6a fall below the 1:1 line. Overall, the above arguments suggest that DOC in the 'lake sediment' layer is subjected to 'active' processing whereas DOC in the 'deeper talik' appears to be more resistant to ongoing and recent degradation.

In Fig. 6b we additionally show that a higher SOM content (LOI) does not automatically result in a more variable $\delta^{13}C_{DOC}$ value range in the 'deeper talik'. Hence, other factors than OC content alone must account for the low degradation rates in the 'deeper talik'. The strong variability in $\delta^{13}C_{DOC}$ values in the 'lake sediment' in contrast to the 'deeper talik' is also well displayed.

De Jong et al. (2018) analyzed the microbial communities in the 'lake sediment' (Unit C) of the same cores analyzed
in this study. They confirmed the presence of a diverse and active microbial community with high specimen numbers including productive methanogens and methanotrophs further emphasizing the observation of a highly active 'lake sediment' layer. This microbial activity may also be methanogenic due to the generally high rates of methane release from Alaskan thermokarst lakes (Elder et al., 2018).

### 4.3 Implications

Based on these findings, we hypothesize that the formation of DOC in the studied thermokarst lake taliks occurs in the individual sedimentary layers without indications of noteworthy DOC exchange between the 'deeper talik' and 'lake sediment'. The main difference is that the 'lake sediment' shows signs of stronger isotopic DOC alteration ('active' DOC) in comparison to the surrounding SOC source material whereas the 'deeper talik' contains DOC with less indication of isotopic alteration ('inactive' DOC) by comparison to its ambient SOC source material (Fig. 7).
Nevertheless, the soil of the 'deeper talik' still experiences more favorable conditions for microbial activity and degradation than the frozen permafrost soil of the surrounding tundra overall. Year-round above-zero conditions in the 'deeper talik' as well as the 'lake sediment' mean both carbon stocks are part of the active carbon cycle. This can ultimately lead to increased GHG emissions from thermokarst lake taliks from both vertical and horizontal talik expansion, especially as the permafrost landscape of the Arctic is currently warming faster than any other region on
the planet (Biskaborn et al., 2019; Box et al., 2019). Changing climatic conditions in an increasingly warming Arctic (Kaufman et al., 2009) with retreating Arctic sea ice (Kinnard et al., 2011) can accelerate thermokarst lake talik formation and degradation in multiple ways. Firstly, increased winter snowfall (Liu et al., 2012) leads to earlier onset





of the spring melt through insulation of the land surface and delays freezing of the lakes impacting maximum winter ice thickness. Secondly, longer, warmer summers and shorter, less cold winters lead to lower maxima in lake ice
thickness. This thinning of maximum winter ice on thermokarst lakes results in a regime shift from bedfast-ice lakes, which freeze down to the lakebed in winter, to floating-ice lakes, that maintain a partially unfrozen water column even in winter. These regime shifts lead to an overall rise in the number and extent of taliks under thermokarst lakes (Arp et al., 2012; Engram et al., 2018). Additionally, ongoing Arctic warming also exacerbates the levels of degradation in thermokarst lake taliks, even in its deeper layers by creating increasingly favorable conditions through
increasing water, soil and sediment temperatures for microbial activity as tested by the temperature sensitivity experiments carried out by de Jong et al. (2018). They showed that a temperature rise from 4 ˚C to 10 ˚C resulted in significantly increased GHG production from thermokarst 'lake sediments'. To what extent, or whether at all, the 'deeper talik' layers would experience stronger degradation and ultimately emit more GHGs is currently unknown and needs to be the subject of further temperature sensitivity studies of 'deeper talik' layers under realistic in-situ
conditions.

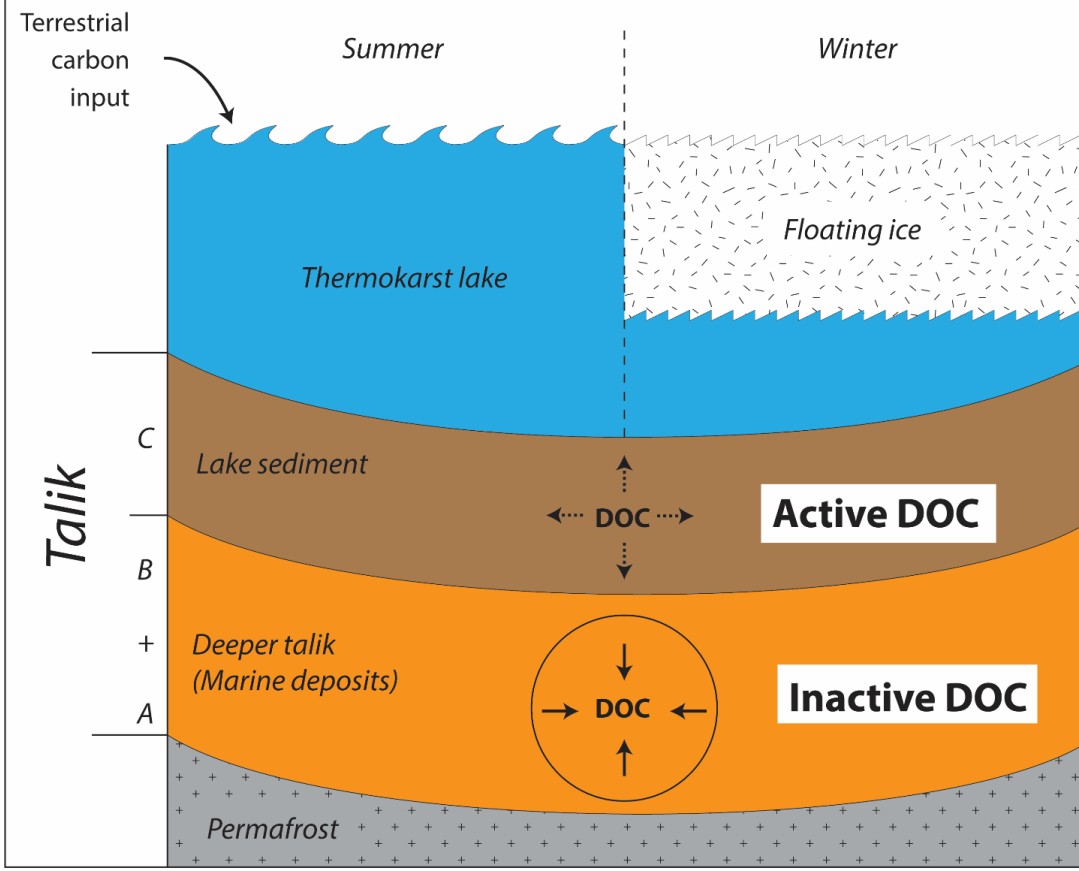

*Figure 7. Conceptual model of DOC dynamics in coastal Alaskan floating-ice thermokarst lake taliks. The uppermost layer of organic-rich lake sediments is an active part of the carbon cycle and appears to undergo high levels of degradation (active DOC). The deeper talik layers of the former permafrost soil also build up high DOC concentrations but appear to be less*
*affected by degradation (inactive DOC).*





## 5 Conclusions

We here used sedimentological and geochemical methods to distinguish and characterize talik layers of different origin below two coastal Alaskan thermokarst lakes. We identified two main units: an organic-rich, young 'lake sediment' of terrestrial origin on top of an older, mineral-rich former permafrost subsoil of marine origin in the 'deeper talik'. We found the 'lake sediment' has a SOC source pool with a narrow $\delta^{13}C$ range and a concurrent $\delta^{13}C_{DOC}$ range that is considerably wider. In contrast, the 'deeper talik' layers have a broad range in $\delta^{13}C_{SOC}$ values but a narrow $\delta^{13}C_{DOC}$ value range. We interpret these opposing trends as an indication for limited levels of DOC degradation in the 'deeper talik' soils and therefore describe its DOC as 'inactive' opposed to the by comparison 'active' DOC in the 'lake sediment' (Fig. 7). Whether these different degrees of degradation in 'deeper talik' vs. 'lake sediment' layers are also expressed as different levels of greenhouse gas emissions is not known. Neither do we know if the level of degradation in 'deeper talik' layers will increase with rising temperatures, along with a potential increase in greenhouse gas production and emissions. We do know that the current rate of lake ice regime shifts from bedfast to floating-ice lakes will have thermal impacts on the underlying soil, and enhanced talik formation will take place in the Arctic during the next decades (Arp et al., 2012). As a result, levels of soil degradation in taliks will overall increase through rising numbers of talik and overall larger talik volumes regardless of the different OC degradation levels in their individual layers.

## Author Contribution

OHM and AJD carried out the field work and sample collection. The XRF analysis and core splitting were carried out by OHM at NIOZ under the supervision of GJR. The samples for the radiocarbon dating were prepared by OHM and analyzed by LW. All other sample preparations, analysis, data collection and processing were carried out by OHM. The interpretation was realized by OHM, JFD, JEV, AJD. The manuscript was written by OHM with contributions from all co-authors.

## Competing Interests

The authors declare that they have no conflict of interest.

## Acknowledgements

This work was carried out under the program of the Netherlands Earth System Science Centre (NESSC). Facilities for magnetic susceptibility, gamma-ray density, TGA and grain size analysis were provided by the Vrije Universiteit Amsterdam, the latter two were supported by Unze van Buuren and Martine Hagen from the sediment laboratory. Thanks to UIC Science, especially Nagruk Harcharek in Utqiaġvik, Alaska for the support during both field trips. The DOC and $\delta^{13}C_{DOC}$ analysis were carried out by Steven Bouillon and Cedric Morana from KU Leuven, Belgium. The $\delta^{13}C$ analysis of the refiltered samples were measured by Richard van Logtestijn at the Department of Systems Ecology of the Vrije Universiteit Amsterdam. The $\delta^{13}C_{SOC}$ analysis were carried out by Suzan Verdegaal-Warmerdam from the Stable Isotope Laboratory of the Vrije Universiteit Amsterdam. Core splitting and XRF scanning of the cores was done at the NIOZ, Texel, Netherlands with the help of Rineke Gieles and Piet van Gaever.



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
