# Peer review of "Porewater $\delta^{13}C_{DOC}$ indicates variable extent of degradation in different talik layers of coastal Alaskan thermokarst lakes"

_Biogeosciences, 2020_

## Referee Comment (RC1) · Anonymous Referee #1 · 4 Jan 2021

This manuscript reads extremely well. The design of the study is described well, the methods are solid, the presentation of the data is very clear. The main finding is that the top sediment layer of thermokarst lakes has more active C cycling than the bottom layer - which stems from a different geological period and has mainly been formed as a marine deposit, and probably prior to the onset of permafrost formation. The results that are discussed most extensively are the carbon isotope data, especially delta-13-C. The discussion section is quite speculative - opening for mechanisms that drive the isotope signal both up and down. It is finally concluded that the relatively large variation of the delta-13-C signal in the top layer is likely to be related to the increased DOM degradation rates.

[Figure]

I would suggest an equally plausible explanation, i.e. that the top layer is more heterogeneous than the bottom layer, because the C inputs here stem from 'old' (permafrost eroded soil) and 'young' (tundra vegetation) sources (how about sphagnum in the ponds? not mentioned - but the ponds are shallow and could have a thriving sphagnum community). The weakness in the data collection is that the isotopic characterisation has not been matched with an assessment of DOM degradability. That could have been done with a simple dark incubation, and would have given a measure of which DOM (talik or lake sediment DOC) was most accessible to microbes. It is likely, of course, that the microbial community in the top is a lot more active (and has access to more nutrients) than in the talik. So some additional nutrient measurements (CN ratios of DOM can also give an indication of DOM nutrient depletion and how far it might be degraded) would have been helpful. So, it seems a bit as though the isotopic data (except for the 14C) actually doesn't help a lot with understanding C cycling in these lakes. I can agree with the conclusion - but not because of the evidence suggested by the 13C results. You also write yourself that the variation in DOC might be source-driven (l 505), but you seem to leave that line of reasoning later in the discussion

So while I admire the manuscript for its clarity and its great explanations of thermokarst formation, I think that the interpretation of the data has some lose ends. 1. I miss a hypothesis (l 98 would be a good place). 2. please add some information about the presence of vegetation / sphagnum in the lakes (minor issue, but good to know). 3. a more extended explanation of the normalizations should be added to the methods. 4. Figure 5a suggests to me that an equal amount of SOM delivers more DOC in the top than the bottom (l 485), which a different interpretation than what you have. Please clarify. 6. Fig 5b (should you turn around the axes? DOC on the x-axis? more intuitive to me), here it looks like the DOC concentration drives the isotope signal and that the isotope signal in both layers is essentially the same, it's just driven by differences in DOC level. This is also shown by figure 6b (LOI and DOC were somewhat interchangeable, weren't they?) for 5a, what is the relationship if you plot it against normalized DOM? 7.

[Figure]

l 505. DOC is not highly bioavailable, especially when it's allochtonous material. It's of course a relative measure, but have a look at the Catalan paper on half-lives of DOM along the aquatic continuum. In boreal systems, oligotrophic, cold, terrigenic DOM can be quite resistant to degradation. 8. Kudos for discussing pH dependence solubility and sorption/desorption of DOM.

---

## Referee Comment (RC2) · Anonymous Referee #2 · 12 Jan 2021

Meisel et al. characterized OC concentrations and degradation beneath two thermokarst lakes in Northern Alaska. They find DOC characteristics originate within individual sediment facies, with little evidence of potential exchange between facies. Using stable isotopes, the authors identify DOC in the lake sediments as showing more evidence of processing compared to the underlying, deeper talik sediments. The findings provide a nice dataset of OC characterization and DOC processing beneath two thermokarst lakes, and their classification of sediments into two primary units (lake sediment and taberites) both fit in with and expand upon previously published research. The paper is well written, and I recommend publication with only a few minor edits.

Minor comments:

The term "layers" should be replaced by "facies," which is the technical term for sediment/geological layers with the same attributes and has been previously used in the context of thermokarst lake sediments (e.g. Farquharson et al., 2016, doi: 10.1016/j.sedgeo.2016.01.002). In thermokarst lake sediment literature, the "deeper talik" facies of former permafrost thawed in situ is referred to as "taberite" (e.g. Grosse et al., 2007, doi: 10.1016/j.geomorph.2006.08.005; Heslop et al., 2015, doi: 10.5194/bg-12-4317-2015; Walter Anthony et al., 2014, doi: 10.1038/nature13560).

Line 17: Remove "up"

Line 62: (Anthony et al., 2014) should be (Walter Anthony et al., 2014)

Figures 2 and 3: The yellow numbers and label, and the light blue "Silicon" label, are hard to read on the white background. Colors with higher contrast would be better.

Figure 4: Same comment as Figures 2 and 3; the yellow is difficult to read on the white background.

---

## Author Comment (AC1) · 16 Feb 2021

**Response to referee comments of manuscript "Porewater $\delta^{13}C_{DOC}$ Indicates Variable Extent Of Degradation In Different Talik Layers Of Coastal Alaskan Thermokarst Lakes"**
**by Ove H. Meisel et al.**

We thank referee #1 for the comments and the constructive discussion of this manuscript. In the following we reply to your remarks. *(Any line number references (L…) are based on the originally uploaded manuscript. Our responses to the comments are written in green color and italic letters. Newly written text that will be added to the revised manuscript is underlined.)*

**Referee Comment #1 ()

This manuscript reads extremely well. The design of the study is described well, the methods are solid, the presentation of the data is very clear. The main finding is that the top sediment layer of thermokarst lakes has more active C cycling than the bottom layer - which stems from a different geological period and has mainly been formed as a marine deposit, and probably prior to the onset of permafrost formation. The results that are discussed most extensively are the carbon isotope data, especially delta-13-C. The discussion section is quite speculative - opening for mechanisms that drive the isotope signal both up and down. It is finally concluded that the relatively large variation of the delta-13-C signal in the top layer is likely to be related to the increased DOM degradation rates. *Dear referee #1, thank you for your positive remarks and constructive comments.*

I would suggest an equally plausible explanation, i.e. that the top layer is more heterogeneous than the bottom layer, because the C inputs here stem from 'old' (permafrost eroded soil) and 'young' (tundra vegetation) sources (how about sphagnum in the ponds? not mentioned - but the ponds are shallow and could have a thriving sphagnum community). The weakness in the data collection is that the isotopic characterization has not been matched with an assessment of DOM degradability. That could have been done with a simple dark incubation, and would have given a measure of which DOM (talik or lake sediment DOC) was most accessible to microbes. It is likely, of course, that the microbial community in the top is a lot more active (and has access to more nutrients) than in the talik. So some additional nutrient measurements (CN ratios of DOM can also give an indication of DOM nutrient depletion and how far it might be degraded) would have been helpful. So, it seems a bit as though the isotopic data (except for the 14C) actually doesn't help a lot with understanding C cycling in these lakes. I can agree with the conclusion - but not because of the evidence suggested by the 13C results. You also write yourself that the variation in DOC might be source-driven (l 505), but you seem to leave that line of reasoning later in the discussion. So while I admire the manuscript for its clarity and its great explanations of thermokarst formation, I think that the interpretation of the data has some lose ends.
*We agree with the above stated main point of criticism that degradation-related data are missing to further back up the observations and conclusions made from the $\delta^{13}C$-based data regarding the different levels of soil degradation in the individual talik layers. We would still like to argue that the conclusions based on the presented $\delta^{13}C$ data were carefully worked out and worded openly to leave room for discussion. But we agree that it is essential to back up the conclusions made in this manuscript with degradation-specific data in future studies. We will add the following section in L545 of the conclusions where we will stress the need for degradation-related data and give examples of possible analysis that would be beneficial.*
*''In future studies this finding needs to be further verified by additional analysis, for example of C:N ratios, dark incubations or humification measurements of the SOM to back up the conclusions based on the $\delta^{13}C_{DOC}$ data and to rule out that the observed trends in carbon isotopes are not solely driven by the sources of sediment but degradation processes.''*
*Nevertheless, we think that the findings are presented with a reasonable level of assumptions. We also see this manuscript as an important gap filler in terms of carbon-related data from these specific Arctic environments where the amount of available data is still scarce.*

*In L496-499 we discuss the possibility of a mainly source-driven $\delta^{13}C_{SOC}$ signal in the lake sediment and deeper talik layers as the input of organic material into the lakes is mainly of terrestrial origin. We point this out to emphasize that $\delta^{13}C_{SOC}$ or $\delta^{13}C_{DOC}$ data alone would not be sufficient to draw conclusions with regard to their degree of OC degradation. But with both data sets in combination the observed changes from $\delta^{13}C_{SOC}$ to $\delta^{13}C_{DOC}$ become interesting indicators of possible degradation activity as they are less likely to be solely source-driven and more likely to display in-situ sediment processes. Please see the above-mentioned sentence we will add to the conclusions section in L545 where we also point out that further analyses are needed to further confirm the results also with regard to the possibility of sediment source-driven isotope signals.*

*Regarding the presence of Sphagnum in the ponds please see our reply below to comment number 2.*

1. I miss a hypothesis (l 98 would be a good place).

*Thank you for pointing this out. Yes, we agree that a clearly stated hypothesis is missing and that L98 is a good place to insert one. We will add the following sentence to L98 to make the intentions of the manuscript clearer to the reader: ''We hypothesize that the level of soil degradation is likely more intense towards the top of the talik in the 'lake sediment' and declines with sediment depth towards the 'deeper talik' layers.''*

2. please add some information about the presence of vegetation / sphagnum in the lakes (minor issue, but good to know).

*For this comment, we would like to point to the below listed parts of the manuscript where the presence of identifiable (where possible) and unidentifiable plant material was discussed in the results section:*
- *L260 (''...clusters of unidentified plant remains...'')*
- *L302-305 (''The sediment of Unit C is for a large part made up by plant detritus of terrestrial origin, mainly in the form of unidentified plant remains but also of roots, small leaves and stems among which brown mosses, Sphagnum and Betula nana were identified.'')*
- *L311-313 (''...Unit C is classified as a homogenous 'lake sediment' that contains large amounts of terrestrial plant detritus...'')*

*Unfortunately, the majority of plant remains were degraded to such a degree that the identification was not possible anymore under a microscope. Therefore, a more in-depth discussion and quantification regarding the presence of different vegetation types found within the talik deposits was not feasible.*

*We agree however, that a short section mentioning the different vegetation types growing in the surrounding landscape and ponds additional to the identified plant remains in the sediment should be added to the manuscript. We will add the following sentence in L130 to the field site section of the results chapter: ''The lakes are surrounded by a marshy tundra landscape with shallow ponds and vegetation consisting of tundra grasses, brown mosses, Sphagnum and occasional Betula nana shrubs.''*

3. a more extended explanation of the normalizations should be added to the methods.

*Agreed. The explanation for the normalization of DOC concentrations and SOM amounts to $DOC_{norm}$ and $SOM_{norm}$, respectively, will be explained in more detail for a better understanding of the applied methods. Please see the new text below which will replace the short description in L341-344 of the manuscript :*

*''The $DOC_{norm}$ data displayed in Fig. 5a are normalized DOC concentrations converted to their total carbon mass in [mg] taking into account the known sediment volume, density and porewater content. The normalization of DOC data points at certain core depths is based on the fixed volume ($V_{core}$) of 1 cm (h) thick cross sections of the sediment core tubes (d = 9 cm) with a volume of 63.6 cm³ ($V_{core} = \pi*(d/2)^2*h$). The total sediment density ($\rho_{sediment\ total}$) of each core section is known from the gamma-ray density measurements. Together with $V_{core}$ the total sediment mass including the pore water ($m_{sediment\ total}$) can be determined for each core depth ($m_{sediment\ total}=V_{core}*\rho_{sediment\ total}$). The wet and dry weights of sediment samples were also measured, thus the relative amount of porewater [wt%] in relation to the total sediment mass (Fig. 2-3) at a certain core depth is also known. Together with $m_{sediment\ total}$ the total porewater mass ($m_{porewater}$) was calculated ($m_{porewater}=m_{sediment\ total}*$porewater amount [wt%]). With $m_{porewater}$ in turn the total porewater volume ($V_{porewater}$) present at a certain sampling depth can be calculated together with the known density of water ($V_{porewater}=m_{porewater}/\rho_{water}$). In a final step $V_{porewater}$ and DOC concentration are used to determine the total mass of DOC present at a certain core depth for that fixed 63.3 cm³ volume ($V_{core}$) of a 1 cm thick core cross*

*section (DOCnorm [mg]=DOC [mg/l] * Vporewater[l]). Normalized DOCnorm values [in mg] display the total amount of DOC present in each individual layer better than relative DOC concentrations [mg/l].''*

4. Figure 5a suggests to me that an equal amount of SOM delivers more DOC in the top than the bottom (l 485), which a different interpretation than what you have. Please clarify.

*In L484-485 we are exclusively talking about the trends in DOC dependence on SOM availability which is in fact very similar and comparable in both layers of lake sediment and the deeper talik. In L476-482 we discuss that the total amount of DOC formed is, as you pointed out as well, in fact higher in the lake sediment in comparison to the deeper talik layer based on Fig. 5a. But we see that the wording in L484-485 can appear ambiguous to the reader and will rephrase it in a clearer way in the revised manuscript:*

*''In Fig. 5a we also show that the formation of porewater DOC is directly linked to the availability of SOM/SOC in the surrounding sediment, as yields of DOC (as DOCnorm) increase with increases in SOMnorm content, although the total amount of DOC yield is significantly higher in the 'lake sediment'.''*

**6.** Fig 5b (should you turn around the axes? DOC on the x-axis? more intuitive to me), here it looks like the DOC concentration drives the isotope signal and that the isotope signal in both layers is essentially the same, it's just driven by differences in DOC level. This is also shown by figure 6b (LOI and DOC were somewhat interchangeable, weren't they?) for 5a, what is the relationship if you plot it against normalized DOM?

*Agreed, we will exchange the horizontal and vertical axes in Fig. 5b of the revised manuscript.*

*Yes, in Fig. 5b the isotope signals of the lake sediment and deeper talik appear to follow the same trend, but it is still clearly visible that the lake sediment has a much broader range in isotopic values than the deeper talik. In Fig. 6b we can show that the isotope data do not follow a clear trend anymore as in Fig. 5b and are therefore not primarily driven by SOM content (L530) which is an important distinction from Fig. 5b.*

*We had not plotted $\delta^{13}C_{DOC}$ against DOCnorm yet but were curious to see how they would correlate (please see graph below). The lake mud (brown) and the deeper talik (blue) are mostly separated in two clusters. While the lake mud still appears to have a trend of increasing carbon isotope values with higher DOC amount the deeper talik appears to be mostly unaffected by changes in DOCnorm. Also, there is no clear trend anymore across the whole sediment core profile ($R^2=0.0115$) as seen in Fig. 5b.*

[Figure]

7. l 505. DOC is not highly bioavailable, especially when it's allochtonous material. It's of course a relative measure, but have a look at the Catalan paper on half-lives of DOM along the aquatic continuum. In boreal systems, oligotrophic, cold, terrigenic DOM can be quite resistant to degradation.

*In our manuscript we argue that the DOC present in the sediment porewater is mainly formed in-situ in the sediment itself and not brought in through hydrological pathways based on the results summarized in Fig. 5a. While the SOC in deeper talik layers and the bulk of the lake sediment are indeed of allochthonous (terrestrial) origin, the porewater DOC appears to be mostly formed within the sediment layers itself. We very much agree that not all DOC compounds are highly bioavailable as we also pointed out in the discussion. We will modify the concerning sentence in L500-501 to make that point about the bioavailability of DOC more clearly:*

*''This assumption is also based on the fact that certain DOC compounds can be  highly bioavailable...''*

*The Catalan et al. (2016) paper also raises the important point of decreasing OC decomposition rates with increased water retention times due to the loss of its most reactive components in the early decomposition processes and an enrichment of the carbon pools with compounds that are degraded more slowly.*

*We also raised the same point in L508-510 of the manuscript: ''Preferential removal of the most labile carbon compounds such as simple polysaccharides which are naturally enriched in $^{13}$C lead to an accumulation in less degradable compounds like lignin...''. In L510 we will add the Catalan et al. (2016) paper as an additional and important reference as well as in the list of references (L659).*

8. Kudos for discussing pH dependence solubility and sorption/desorption of DOM.

*Thank you for the kind acknowledgement.*

---

## Author Comment (AC2) · 16 Feb 2021

**Response to referee comments of manuscript "Porewater $\delta^{13}C_{DOC}$ Indicates Variable Extent Of Degradation In Different Talik Layers Of Coastal Alaskan Thermokarst Lakes"**
**by Ove H. Meisel et al.**

We thank referee #2 for the comments and the constructive discussion of this manuscript. In the following we reply to your remarks. *(Any line number references (L…) are based on the originally uploaded manuscript. Our responses to the comments are written in green color and italic letters. Newly written text that will be added to the revised manuscript is underlined.)*

**Referee Comment #2** ()
Meisel et al. characterized OC concentrations and degradation beneath two thermokarst lakes in Northern Alaska. They find DOC characteristics originate within individual sediment facies, with little evidence of potential exchange between facies. Using stable isotopes, the authors identify DOC in the lake sediments as showing more evidence of processing compared to the underlying, deeper talik sediments. The findings provide a nice dataset of OC characterization and DOC processing beneath two thermokarst lakes, and their classification of sediments into two primary units (lake sediment and taberites) both fit in with and expand upon previously published research. The paper is well written, and I recommend publication with only a few minor edits.
*Dear referee #2, we thank you for your positive assessment and kind words.*

Minor comments:
The term "layers" should be replaced by "facies," which is the technical term for sediment/geological layers with the same attributes and has been previously used in the context of thermokarst lake sediments (e.g. Farquharson et al., 2016, doi: 10.1016/j.sedgeo.2016.01.002).
*We agree that the term "facies'' is more suitable when the text refers specifically to sedimentary/geological units. The word "layer" will be replaced (40 times) by the technical term "facies'' in most of the manuscript (L97a, L97b, L100, L101, L103, L250, L251a, L251b, L253, L257, L314, L333, L344, L357, L385, L429, L472, L473, L474, L481, L484, L485, L488, L490, L491a, L491b, L498, L518, L528, L537, L541, L559, L563, L564, L567, L569, L577, L581, L582, L587). In several instances we will leave the term "layer" which allows for a better flow of reading especially when the text explicitly refers to talik units in general and not specifically to sedimentary/geological units (L95a, L95b, L129, L149, L262, L437). This also includes the title (L2) and the whole abstract (L21, L 23, L27, L40) of the manuscript. We believe that it will be easier for a broader range of readers from different scientific fields to follow the manuscript when the wording of the title and abstract are kept in more general terminologies. The term "facies'' will be introduced in L96 where it will be used together with the term "layer", additionally, the suggested reference (Farquharson et al., 2016) will be added in L96 and in the list of references (L681). In the conclusions in L573 the terms "layer" and "facies'' will be also be used together for a better understanding for readers who quickly screen through the abstract and the conclusions first before reading the complete manuscript.*

In thermokarst lake sediment literature, the "deeper talik" facies of former permafrost thawed in situ is referred to as "taberite" (e.g. Grosse et al., 2007, doi: 10.1016/j.geomorph.2006.08.005; Heslop et al., 2015, doi: 10.5194/bg-12-4317-2015; Walter Anthony et al., 2014, doi: 10.1038/nature13560).
*We will replace the descriptive term 'deeper talik' in the whole manuscript (71 times) and in Fig. 5, 6 and 7 (3 times) with the scientific term 'taberite', strictly based on the definition of the word in the suggested references (Grosse et al., 2007; Heslop et al., 2015; Walter Anthony et al., 2014). The term 'taberite' will be further explained in L79 of the manuscript with referral to the above mentioned publications:*
*''In particular, it is unclear to what degree the uppermost talik layer of the 'lake sediment' and the thawed permafrost soil layer, known as taberite (Grosse et al., 2007; Walter Anthony et al., 2014), are affected by degradation processes (Heslop et al., 2015).''*
*(The publication by Grosse et al., 2007 will also be added to the list of references in L681.)*

Line 17: Remove "up"
*Agreed, the word 'up' in L17 will be removed in the revised manuscript.*

Line 62: (Anthony et al., 2014) should be (Walter Anthony et al., 2014)
*Thank you for pointing this out. The reference will be adapted accordingly in the manuscript in L62 and L66 and in the list of references (L620).*

Figures 2 and 3: The yellow numbers and label, and the light blue "Silicon" label, are hard to read on the white background. Colors with higher contrast would be better. Figure 4: Same comment as Figures 2 and 3; the yellow is difficult to read on the white background.
*Agreed, the yellow/orange and light blue colors are hard to distinguish from the background in all three figures. We will change the axis labeling, tick mark labeling and the plot line color in Fig. 2, 3 and 4 to higher contrast colors for better visualization.*

---

## Author Response (AR1)

**Author's Response of the manuscript "Porewater $\delta^{13}C_{DOC}$ indicates variable extent of degradation in different talik layers of coastal Alaskan thermokarst lakes" by Ove H. Meisel et al.**

We thank both referees and the editor Yakov Kuzyakov for their comments and the constructive discussion of this manuscript. In the following we reply to your remarks. *(Any line number references (L…) are based on the revised manuscript. Our responses to the comments are written in green color and italic letters. Newly written text that will be added to the revised manuscript is underlined.)*

**Referee Comment #1** ()

This manuscript reads extremely well. The design of the study is described well, the methods are solid, the presentation of the data is very clear. The main finding is that the top sediment layer of thermokarst lakes has more active C cycling than the bottom layer - which stems from a different geological period and has mainly been formed as a marine deposit, and probably prior to the onset of permafrost formation. The results that are discussed most extensively are the carbon isotope data, especially delta-13-C. The discussion section is quite speculative - opening for mechanisms that drive the isotope signal both up and down. It is finally concluded that the relatively large variation of the delta-13-C signal in the top layer is likely to be related to the increased DOM degradation rates. *Dear referee#1, thank you for your positive remarks and constructive comments.*

I would suggest an equally plausible explanation, i.e. that the top layer is more heterogeneous than the bottom layer, because the C inputs here stem from 'old' (permafrost eroded soil) and 'young' (tundra vegetation) sources (how about sphagnum in the ponds? not mentioned - but the ponds are shallow and could have a thriving sphagnum community). The weakness in the data collection is that the isotopic characterization has not been matched with an assessment of DOM degradability. That could have been done with a simple dark incubation, and would have given a measure of which DOM (talik or lake sediment DOC) was most accessible to microbes. It is likely, of course, that the microbial community in the top is a lot more active (and has access to more nutrients) than in the talik. So some additional nutrient measurements (CN ratios of DOM can also give an indication of DOM nutrient depletion and how far it might be degraded) would have been helpful. So, it seems a bit as though the isotopic data (except for the 14C) actually doesn't help a lot with understanding C cycling in these lakes. I can agree with the conclusion - but not because of the evidence suggested by the 13C results. You also write yourself that the variation in DOC might be source-driven (l 505), but you seem to leave that line of reasoning later in the discussion. So while I admire the manuscript for its clarity and its great explanations of thermokarst formation, I think that the interpretation of the data has some lose ends.

*We agree with the above stated main point of criticism that degradation-related data are missing to further back up the observations and conclusions made from the $\delta^{13}C$-based data regarding the different levels of soil degradation in the individual talik layers. We would still like to argue that the conclusions based on the presented $\delta^{13}C$ data were carefully worked out and worded openly to leave room for discussion. But we agree that it is essential to back up the conclusions made in this manuscript with degradation-specific data in future studies. We added a section in L601-604 of the conclusions where we stress the need for degradation-related data and give examples of possible analysis that would be beneficial.*
*Nevertheless, we think that the findings are presented with a reasonable level of assumptions. We also see this manuscript as an important gap filler in terms of carbon-related data from these specific Arctic environments where the amount of available data is still scarce.*

*In L517-523 we discuss the possibility of a mainly source-driven $\delta^{13}C_{SOC}$ signal in the lake sediment and deeper talik layers as the input of organic material into the lakes is mainly of terrestrial origin. We point this out to emphasize that $\delta^{13}C_{SOC}$ or $\delta^{13}C_{DOC}$ data alone would not be sufficient to draw conclusions with regard to their degree of OC degradation. But with both data sets in combination the observed changes from $\delta^{13}C_{SOC}$ to $\delta^{13}C_{DOC}$ become interesting indicators of possible degradation activity as they are less likely to be solely source-driven and more*

*likely to display in situ sediment processes. Please see the below-mentioned sentence we that was added to the conclusions section in L601-604 where we also point out that further analyses are needed to further confirm the results also with regard to the possibility of sediment source-driven isotope signals.*

*Regarding the presence of Sphagnum in the ponds please see our reply to comment number 2.*

*L601-604:*
*'' In future studies this finding needs to be further verified by additional analysis, for example of C:N ratios, dark incubations or humification measurements of the SOM to back up the conclusions based on the $\delta^{13}C_{DOC}$ data and to rule out that the observed trends in carbon isotopes are not solely driven by the sources of sediment but degradation processes.''*

1. I miss a hypothesis (l 98 would be a good place).
*Thank you for pointing this out. Yes, we agree that a clearly stated hypothesis is missing and that L98 is a good place to insert one. We added a sentence with a hypothesis to L98-100 to make the intentions of the manuscript clearer to the reader.*

*L98-100:*
*''We hypothesize that the level of soil degradation is likely more intense towards the top of the talik in the 'lake sediment' and declines with sediment depth towards the 'taberite' facies.''*

2. please add some information about the presence of vegetation / sphagnum in the lakes (minor issue, but good to know).
*For this comment, we would like to point to the below listed parts of the manuscript where the presence of identifiable (where possible) and unidentifiable plant material was discussed in the results section:*
   - *L266 (''...clusters of unidentified plant remains...'')*
   - *L306-304 (''The sediment of Unit C is for a large part made up by plant detritus of terrestrial origin, mainly in the form of unidentified plant remains but also of roots, small leaves and stems among which brown mosses, Sphagnum and Betula nana were identified.'')*
   - *L316-317 (''...Unit C is classified as a homogenous 'lake sediment' that contains large amounts of terrestrial plant detritus...'')*
*Unfortunately, the majority of plant remains were degraded to such a degree that the identification was not possible anymore under a microscope. Therefore, a more in-depth discussion and quantification regarding the presence of different vegetation types found within the talik deposits was not feasible.*
*We agree however, that a short section mentioning the different vegetation types growing in the surrounding landscape and ponds additional to the identified plant remains in the sediment should be added to the manuscript. Therefore, we added an additional sentence in L133-135 to the field site section of the results chapter.*

*L133-135:*
*''The lakes are surrounded by a marshy tundra landscape with shallow ponds and vegetation consisting of tundra grasses, brown mosses, Sphagnum and occasional Betula nana shrubs.''*

3. a more extended explanation of the normalizations should be added to the methods.
*Agreed. The explanation for the normalization of DOC concentrations and SOM amounts to $DOC_{norm}$ and $SOM_{norm}$, respectively, will be explained in more detail for a better understanding of the applied methods. Please see the new text below which replaces the previously short description in the manuscript.*

*L339-357 & 364-365:*
*''The $DOC_{norm}$ data displayed in Fig. 5a are normalized DOC concentrations converted to their total carbon mass in [mg] taking into account the known sediment volume, density and porewater content. The normalization of DOC data points at certain core depths is based on the fixed volume ($V_{core}$) of 1 cm (h) thick cross sections of the sediment core tubes (d = 9 cm) with a volume of 63.6 cm³.*

$$V_{core} = \pi \cdot \left(\frac{d}{2}\right)^2 \cdot h \qquad\qquad\qquad (1)$$

*The total sediment density ($\rho_{sediment\ total}$) of each core section is known from the gamma-ray density measurements. Together with $V_{core}$ the total sediment mass including the pore water ($m_{sediment\ total}$) can be determined for each core depth.*

$$m_{sediment\ total} = V_{core} \cdot \rho_{sediment\ total} \qquad\qquad\qquad (2)$$

*The wet and dry weights of sediment samples were also measured, thus the relative amount of porewater [wt%] in relation to the total sediment mass (Fig. 2-3) at a certain core depth is also known. Together with $m_{sediment\ total}$ the total porewater mass ($m_{porewater}$) was calculated.*

$$m_{porewater} = m_{sediment\ total} \cdot porewater\ [wt\%] \qquad\qquad\qquad (3)$$

*With $m_{porewater}$ in turn the total porewater volume ($V_{porewater}$) present at a certain sampling depth can be calculated together with the known density of water ($\rho_{water}$).*

$$V_{porewater} = \frac{m_{porewater}}{\rho_{water}} \qquad\qquad\qquad (4)$$

*In a final step $V_{porewater}$ and DOC concentration are used to determine the total mass of DOC present at a certain core depth for that fixed 63.3 cm³ volume ($V_{core}$) of a 1 cm thick core cross section:*

$$DOC_{norm}\ [mg] = DOC\ [mg\ l^{-1}] \cdot V_{porewater}\ [l] \qquad\qquad\qquad (5)$$

*Normalized $DOC_{norm}$ values [in mg] display the total amount of DOC present in each individual layer better than relative DOC concentrations [mg/l].''*

4. Figure 5a suggests to me that an equal amount of SOM delivers more DOC in the top than the bottom (l 485[*now 504-505*]), which a different interpretation than what you have. Please clarify.
*In L504-505 we are exclusively talking about the trends in DOC dependence on SOM availability which is in fact very similar and comparable in both layers of lake sediment and the deeper talik. In L494-496 we discuss that the total amount of DOC formed is, as you pointed out as well, in fact higher in the lake sediment in comparison to the deeper talik layer based on Fig. 5a. But we see that the wording in L497-499 can appear ambiguous to the reader and rephrased it in a clearer way in the revised manuscript.*

*L497-499:*
*''In Fig. 5a we also show that the formation of porewater DOC is directly linked to the availability of SOM/SOC in the surrounding sediment, as yields of DOC (as $DOC_{norm}$) increase with increases in $SOM_{norm}$ content, although the total amount of DOC yield is significantly higher in the 'lake sediment'.''*

**6.** Fig 5b (should you turn around the axes? DOC on the x-axis? more intuitive to me), here it looks like the DOC concentration drives the isotope signal and that the isotope signal in both layers is essentially the same, it's just driven by differences in DOC level. This is also shown by figure 6b (LOI and DOC were somewhat interchangeable, weren't they?) for 5a, what is the relationship if you plot it against normalized DOM?
*Agreed, we will exchange the horizontal and vertical axes in Fig. 5b of the revised manuscript.*
*Yes, in Fig. 5b the isotope signals of the lake sediment and deeper talik appear to follow the same trend, but it is still clearly visible that the lake sediment has a much broader range in isotopic values than the deeper talik. In Fig. 6b we can show that the isotope data do not follow a clear trend anymore as in Fig. 5b and are therefore not primarily driven by SOM content (L551-554) which is an important distinction from Fig. 5b.*
*We had not plotted $\delta^{13}C_{DOC}$ against $DOC_{norm}$ yet but were curious to see how they would correlate (please see graph below). The lake mud (brown) and the deeper talik (blue) are mostly separated in two clusters. While the lake mud still appears to have a trend of increasing carbon isotope values with higher DOC amount the deeper talik appears*

*to be mostly unaffected by changes in DOC$_{norm}$. Also, there is no clear trend anymore across the whole sediment core profile ($R^2$=0.0115) as seen in Fig. 5b.*

[Figure]

*The horizontal and vertical axes were exchanged including the axis labeling. The title of the graph was accordingly also changed from 'DOC/$\delta^{13}C_{DOC}$' to '$\delta^{13}C_{DOC}/DOC$'.*

7. l 505 [*now 521*]. DOC is not highly bioavailable, especially when it's allochtonous material. It's of course a relative measure, but have a look at the Catalan paper on half-lives of DOM along the aquatic continuum. In boreal systems, oligotrophic, cold, terrigenic DOM can be quite resistant to degradation.

*In our manuscript we argue that the DOC present in the sediment porewater is mainly formed in-situ in the sediment itself and not brought in through hydrological pathways based on the results summarized in Fig. 5a. While the SOC in deeper talik layers and the bulk of the lake sediment are indeed of allochthonous (terrestrial) origin, the porewater DOC appears to be mostly formed within the sediment layers itself. We very much agree that not all DOC compounds are highly bioavailable as we also pointed out in the discussion. We modified the concerning sentence in L520-523 to make that point about the bioavailability of DOC more clearly.*

*The Catalan et al. (2016) paper also raises the important point of decreasing OC decomposition rates with increased water retention times due to the loss of its most reactive components in the early decomposition processes and an enrichment of the carbon pools with compounds that are degraded more slowly.*
*We also raised the same point in L528-530 of the manuscript: ''Preferential removal of the most labile carbon compounds such as simple polysaccharides which are naturally enriched in $^{13}C$ lead to an accumulation in less degradable compounds like lignin…''. In L530 we, therefore, added the Catalan et al. (2016) paper as an additional and important reference as well as in the list of references (L682-683).*

*L520-523:*
*''This assumption is also based on the fact that certain DOC compounds can be  highly bioavailable, as  they  are quickly consumed by microorganisms and transformed by biochemical degradation (Battin et al., 2008) and thereby making  changes from $\delta^{13}C_{SOC}$ to $\delta^{13}C_{DOC}$ a more sensitive indicator for the degree of OC degradation than $\delta^{13}C_{SOC}$ or $\delta^{13}C_{DOC}$ by itself.''*

*L530:*
*We added the new reference 'Catalán et al. (2016)' in the text and the full reference in the list of references (L682-683): Catalán, N., Marcé, R., Kothawala, D. N. and Tranvik, L. J.: Organic carbon decomposition rates controlled by water retention time across inland waters, Nat. Geosci., 9(7), 501–504, doi:10.1038/ngeo2720, 2016.*

8. Kudos for discussing pH dependence solubility and sorption/desorption of DOM.
*Thank you for the kind acknowledgement.*

**Referee Comment #2** ()
Meisel et al. characterized OC concentrations and degradation beneath two thermokarst lakes in Northern Alaska. They find DOC characteristics originate within individual sediment facies, with little evidence of potential exchange between facies. Using stable isotopes, the authors identify DOC in the lake sediments as showing more evidence of processing compared to the underlying, deeper talik sediments. The findings provide a nice dataset of OC characterization and DOC processing beneath two thermokarst lakes, and their classification of sediments into two primary units (lake sediment and taberites) both fit in with and expand upon previously published research. The paper is well written, and I recommend publication with only a few minor edits.
*Dear referee#2, we thank you for your positive assessment and kind words.*

Minor comments:
The term "layers" should be replaced by "facies," which is the technical term for sediment/geological layers with the same attributes and has been previously used in the context of thermokarst lake sediments (e.g. Farquharson et al., 2016, doi: 10.1016/j.sedgeo.2016.01.002).
*Dear referee#2, we agree that the term "facies'' is more suitable when the text refers specifically to sedimentary/geological units. The word "layer" is now replaced (43 times) by the technical term "facies'' in most of the manuscript. In several instances we left the term "layer" which allows for a better flow of reading especially when the text explicitly refers to talik units in general and not specifically to sedimentary/geological units (L95a, L95b, L133, L154, L278, L364, L456). This also includes the title (L2) and the whole abstract (L21, L 24, L27, L39) of the manuscript. We believe that it is easier for a broader range of readers from different scientific fields to follow the manuscript when the wording of the title and abstract are kept in more general terminologies. The term "facies'' is now introduced in L96 where it is used together with the term "layer", additionally, the suggested reference (Farquharson et al., 2016) was added in L96 and in the list of references (L706-708). In the conclusions in L594 the terms "layer" and "facies'' were also used together for a better understanding for readers who quickly screen through the abstract and the conclusions first before reading the complete manuscript.*

*L96 and L681:*
*We introduced the term 'facies' and added the new reference 'Farquharson et al. (2016)' in L96. The list of references (L706-708) was updated accordingly:*
*Farquharson, L., Walter Anthony, K. W., Bigelow, N., Edwards, M. and Grosse, G.: Facies analysis of yedoma thermokarst lakes on the northern Seward Peninsula, Alaska, Sediment. Geol., 340, 25–37, doi:10.1016/j.sedgeo.2016.01.002, 2016.*

In thermokarst lake sediment literature, the "deeper talik" facies of former permafrost thawed in situ is referred to as "taberite" (e.g. Grosse et al., 2007, doi: 10.1016/j.geomorph.2006.08.005; Heslop et al., 2015, doi: 10.5194/bg-12-4317-2015; Walter Anthony et al., 2014, doi: 10.1038/nature13560).
*We replaced the descriptive term 'deeper talik' in the whole manuscript (75 times) and in Fig. 5, 6 and 7 (3 times) with the scientific term 'taberite', strictly based on the definition of the word in the suggested references (Grosse et al., 2007; Heslop et al., 2015; Walter Anthony et al., 2014). The term 'taberite' is now further explained in L24 of the abstract and in L78-79 of the manuscript with referral to the above mentioned publications.*
*(The publication by Grosse et al., 2007 was also added to the list of references in L709-711.)*

*L78-79:*
*''The deeper part of the talik contains the previously frozen ground of the permafrost underlying the lake, these thawed permafrost soils are called 'taberite' (Grosse et al, 2007; Walter Anthony et al., 2014; Heslop et al., 2015).''*

*L709-711:*
*New reference 'Grosse et al. (2007)' was added to the list of references:*
*Grosse, G., Schirrmeister, L., Siegert, C., Kunitsky, V. V., Slagoda, E. A., Andreev, A. A. and Dereviagyn, A. Y.: Geological and geomorphological evolution of a sedimentary periglacial landscape in Northeast Siberia during the Late Quaternary, Geomorphology, 86(1–2), 25–51, doi:10.1016/j.geomorph.2006.08.005, 2007.*

*Fig. 5a, 6a and 7 (L374, L403, L587):*
*The term 'deeper talik' was replaced by the term 'taberite' in these three figures.*

Line 17: Remove "up"
*Agreed, the word 'up' in L17 will is now removed in the revised manuscript.*

*L17:*
*Removed the word ''.*

Line 62: (Anthony et al., 2014) should be (Walter Anthony et al., 2014)
*Thank you for pointing this out. The reference is now adapted accordingly in the manuscript in L61, L65 and the list of references where it is now moved to L620.*

*L61, L65 and L817:*
*Changed reference ''(Anthony et al., 2014)'' to ''(Walter Anthony et al., 2014)''.*

Figures 2 and 3: The yellow numbers and label, and the light blue "Silicon" label, are hard to read on the white background. Colors with higher contrast would be better. Figure 4: Same comment as Figures 2 and 3; the yellow is difficult to read on the white background.
*Agreed, the yellow/orange and light blue colors are hard to distinguish from the background in all three figures. We changed the axis labeling, tick mark labeling and the plot line color in Fig. 2, 3 and 4 to higher contrast colors for better visualization.*

*Fig. 2, 3 (L268, 273):*
*The color of the axis labels, tick marks and plotlines for silicon (Si) were changed from a light blue to a darker blue.*
*The color of the axis labels, tick marks and plotlines for density were changed from orange to a darker brown.*

*Fig. 4 (L358):*
*The color of the axis labels and plotlines for DOC were changed from a light blue to a dark blue.*
*The color of the axis labels and plotlines for $\delta^{13}C_{SOC}$ were changed from orange to a darker brown.*

**Additional Text Changes**

*Dear Editor, we would appreciate it if the ORCIDs of all contributing authors would be added to the manuscript:*

*ORCIDs:*
*Ove H. Meisel:*           *0000-0002-6431-6659*
*Joshua F. Dean:*          *0000-0001-9058-7076*
*Jorien E. Vonk:*          *0000-0002-1206-5878*
*Lukas Wacker:*           *0000-0002-8215-2678*
*Gert-Jan Reichart:*        *0000-0002-7256-2243*
*Han A.J. Dolman:*         *0000-0003-0099-0457*

*- Spaces in between ranges (e.g. U1-U3) were deleted in the complete manuscript.*

*- Spaces in between temperature values and units were added in the complete manuscript (e.g. 550 °C).*

*- Changed all units with a denominator to negative exponents (for example [mg/l] to [mg l$^{-1}$] or [g/cm³] to [g cm$^{-3}$]).*

*L1-2:*
*Changed capital letters in title from '' Porewater $\delta^{13}C_{DOC}$ Indicates Variable Extent Of Degradation In Different Talik Layers Of Coastal Alaskan Thermokarst Lakes'' to '' Porewater $\delta^{13}C_{DOC}$ indicates variable extent of degradation in different talik layers of coastal Alaskan thermokarst lakes''*

*L160:*
*Changed ''…6 hours…'' to ''…six hours…''.*

*L166-167:*
*Changed ''…3-4 drops…'' to ''…three to four drops…''.*

*L169:*
*Changed ''…2 weeks…'' to ''…two weeks…''.*

*L246 & L247:*
*Changed ''…1 hour…'' to ''…one hour…''.*

*L318:*
*Added 'the' to '' …facies of the Units A/B.''.*

*L376:*
*Changed 'pore water' to 'porewater'.*

*L409:*
*Changed ''soil organic matter (SOM)'' to ''SOM''.*

*L410:*
*Changed '' The LOI data (equivalent for SOM content)…'' to '' The LOI data (as equivalent for SOM content)…''*

*L437:*
*Moved ''(Table 2)'' to L436.*

*L528:*
*Changed 'δ13C' to 'δ$^{13}$C'.*

*L502, 509, 540, 586:*
*Changed 'in-situ' to 'in situ'.*

*L569, L583, L585:*
*Exchanged 'GHG(s)' for 'greenhouse gas(es)' (unnecessary abbreviation in the manuscript).*

*L596:*
*Changed '' ...of terrestrial origin…'' to '' ...of mostly terrestrial origin…''.*
*L603:*
*Changed ''...numbers of talik…'' to ''...numbers of taliks..''.*

*L620:*
*Added data availibility section:*

*''**Data Availability***

*The data presented in this study are available on DataverseNL at https://doi.org/10.34894/XK4LSU. ''*

*L627:*
*Added 'and funded' to '' This work was carried out and funded under the program of…'''.*

**Additional Figure Changes**

*Fig. 1 (L104):*
*- Units for lon and lat were adapted in the small Alaska overview map. (e.g. '60°N' was changed to '60° N' etc.).*

*Fig. 2, 3 (L268 & L273):*
*- Axis titles for 'Pore Water [wt%]' were changed to 'Porewater [wt%]'.*
*- Plotlines and plotline examples in axis labels for density were changed from a solid line to a dashed line.*
*- Colors of bulk sediment radiocarbon ages on top of core E3 (Fig. 2) and core U2 (Fig. 3) were changed from a light blue to a higher contrast dark blue.*
*- Axes unit labeling for magnetic susceptibility was changed from [cm³/g] to [cm³ g$^{-1}$].*
*- Axes unit labeling for density was changed from [g/cm³] to [g cm$^{-3}$].*

*Fig. 4 (L358):*
*- The $\delta^{13}C_{SOC}$ plotline and axis label example were changed from a solid to a dashed line.*
*- The $\delta^{13}C_{DOC}$ plotline and axis label example were changed from a dashed to a solid line.*
*- Axis unit labeling for DOC was changed from [mg/l] to [mg l$^{-1}$].*

*Fig. 5b (L374):*
*- Axis unit labeling for DOC was changed from [mg/l] to [mg l$^{-1}$].*